# The dispersal of fluvially discharged and marine, shelf-produced particulate organic matter in the northern Gulf of Mexico

Yord W. Yedema[1], Francesca Sangiorgi[1], Appy Sluijs[1], Jaap S. Sinninghe Damsté[1,2], Francien Peterse[1]

[1] Department of Earth Sciences, Utrecht University, 3584 CB Utrecht, the Netherlands
[2]Department of Marine Microbiology and Biogeochemistry, NIOZ Royal Netherlands Institute for Sea Research, Den Burg, the Netherlands

*Correspondence to*: Yord W. Yedema (y.w.ijedema@uu.nl)

**Abstract.** Rivers play a key role in the global carbon cycle by transporting terrestrial organic matter (TerrOM) from land to the ocean. Upon burial in marine sediments, this TerrOM may be a significant long-term carbon sink, depending on its composition and properties. However, much remains unknown about the dispersal of different types of TerrOM in the marine realm upon fluvial discharge as the commonly used bulk OM parameters do not reach the required level of source- and process-specific information. Here, we analysed bulk OM properties, lipid biomarkers (long-chain *n*-alkanes, sterols, long-chain diols, alkenones, branched and isoprenoid glycerol dialkyl glycerol tetraethers (GDGTs)), pollen, and dinoflagellate cysts in marine surface sediments along two transects offshore the Mississippi and Atchafalaya Rivers (MAR), as well as one along the 20 m isobath in the direction of the river plume. We use these biomarkers and palynological proxies to identify the dispersal patterns of soil-microbial- (SMOM), fluvial, higher plant, and marine produced OM in the coastal sediments of the northern Gulf of Mexico (GoM).

The Branched and Isoprenoid Tetraether (BIT) index and the relative abundance of $C_{32}$ 1,15-diols indicative for freshwater production show high contributions of SMOM and fluvial OM near the Mississippi River mouth (BIT = 0.6, $F_{C32\ 1,15}$ >50%), which rapidly decrease further away from the river mouth (BIT <0.1, $F_{C32\ 1,15}$ <20%). In contrast, concentrations of long-chain *n*-alkanes and pollen grains do not show this stark decrease along the path of transport, and especially *n*-alkanes are also found in sediments in deeper waters. Proxy indicators show that marine productivity is highest close to shore, and reveal that marine producers (diatoms, dinoflagellates, coccolithophores) have different spatial distributions, indicating their preferred niches. Close to the coast, where food supply is high and waters are turbid, cysts of heterotrophic dinoflagellates dominate the assemblages. The dominance of heterotrophic taxa in shelf waters in combination with the rapid decrease in the relative contribution of TerrOM towards the deeper ocean, suggests that TerrOM input may trigger a priming effect that results in its rapid decomposition upon discharge. In the open ocean far away from the river plume, autotrophic dinoflagellates dominate the assemblages, indicating more oligotrophic conditions.

Our combined lipid biomarker and palynology approach reveals that different types of TerrOM have distinct dispersal patterns, suggesting that the initial composition of this particulate OM influences the burial efficiency of TerrOM on the continental margin.

## 1 Introduction

The transport of terrestrial organic matter (TerrOM) from land to the sea, and subsequent discharge into the (open) ocean is an important process that connects the terrestrial and marine carbon cycles (Blair & Aller, 2012). This process is mostly conducted by rivers. Most TerrOM that enters the river originates from soil mobilisation and weathering of rocks and includes plant-derived, soil microbial- (SMOM), and petrogenic OM, as well as aquatic OM produced in lakes and streams. In rivers, TerrOM can be transported as particulate organic carbon (POM) or dissolved organic carbon (DOM), where POM often includes old, degraded plant and soil material, while DOM is generally younger (Mayorga et al., 2005). During river transport, partial degradation and/or (temporal) storage in sediments and inland wetlands can change the composition of TerrOM (Battin et al.,

2009; Aufdenkampe et al., 2011). Therefore, only a fraction of the OM delivered to rivers (32-47%) reaches the coastal zone (Cole et al., 2007; Battin et al., 2009; Tranvik et al., 2009; Aufdenkampe et al., 2011; Regnier et al., 2013; Li et al., 2017; Kirschbaum et al., 2019).

After its arrival in the ocean, TerrOM may be buried in marine sediments, where it can form a long-term sink of atmospheric $CO_2$ (Hedges et al., 1997), but burial rates in coastal zones are not homogeneous. Regions that receive substantial input of OM, such as river-dominated continental shelves, form potential hotspots for OM burial (e.g., Bianchi et al., 2018). On a global scale, it is estimated that only about one-third of the fluvially discharged POM is eventually preserved in marine sediments (Burdige, 2005; Blair and Aller, 2012), indicating that part of the TerrOM is lost in the marine realm. The dispersal of TerrOM in the marine realm depends on many factors, including its molecular composition, the mode of transport, the recalcitrance/ degradability of TerrOM and oxygen exposure time (e.g., Zonneveld et al., 2010). One component that is often overlooked in marine sediments is the relative contribution of different sources of TerrOM (i.e., microbial, plant, aquatic) that make up its composition. Since the contribution of TerrOM to marine sediments is often investigated using bulk parameters only, the composition of TerrOM is not generally characterized, whereas the different components may have a distinct dispersal patterns and burial efficiencies. In addition, TerrOM contributions to marine sediments might be masked if only bulk parameters are used, thereby influencing our understanding of TerrOM dispersal patterns in the marine environment (e.g., Goñi et al., 1997; 1998).

The northern Gulf of Mexico (GoM) receives high inputs of OM from the Mississippi-Atchafalaya River (MAR) system. As the MAR catchment area covers a large part of North America, it transports a large variety of TerrOM types from the continent to the ocean, making the GoM an interesting region to investigate the spatial distribution of different TerrOM pools. Using primary productivity rates, TOC and comparisons with $\delta^{13}C_{org}$ values, Trefry et al. (1994) estimated that ~20-50% of the particulate organic carbon (POC) flux off the Mississippi river is buried on the Louisiana shelf and that <40% is terrestrially derived. Studies on organic matter sources in the northern GoM generally focussed on total OM concentrations, or the differentiation of terrestrial and marine OM sources, mostly using bulk OM properties (TOC and $\delta^{13}C_{org}$ values) and concentrations/ratios of plant (lignin) and/or algal (photosynthetic pigments) biomarkers (e.g., Goñi et al., 1997; 1998; Bianchi et al., 2002; Chen et al., 2003; Wysocki et al., 2006; Waterson and Canuel, 2008; Sampere et al., 2008; 2011). These studies showed that TerrOM concentrations follow the expected trend of highest concentrations close to the MR and a subsequent decrease with increasing distance from the river mouth. In addition, the composition of sedimentary OM was found to gradually change from more plant-derived OM close to the coast (Bianchi et al., 2002; Sampere et al., 2008) to more algal-derived OM along the shelf and further downslope (Chen et al., 2003; Wysocki et al., 2006). Regardless, TerrOM was still present in sediments from the most distal locations that were studied (>2000 m water depth, Goñi et al., 1997).

Dispersal patterns of different types of TerrOM have previously been studied with ratios based on lignin phenols (Goñi et al., 1997; 1998; Bianchi et al., 2002; Sampere et al., 2008). While these studies have demonstrated that different TerrOM pools do have specific dispersal patterns, little is known about the source compositions of the TerrOM that is buried in the northern GoM sediments, as lignin phenols only represents plant-derived OM. Here, we here aim to disentangle the source(s) of TerrOM

in 20 marine surface sediments offshore the Mississippi and Atchafalaya River delta (Fig. 1) using lipid biomarkers and palynological methods (dinoflagellate cysts, pollen) that are specific for SMOM, fluvial-, plant- and marine OM, supported by bulk parameters (total organic carbon (TOC), the organic carbon/total nitrogen ratio (C/N ratio) and bulk $\delta^{13}C_{org}$) of the sediments. In addition, the marine biomarkers are used to identify the preferred niches of different OM producers in the GoM and assess their possible relation to the TerrOM distribution. This combined biomarker and palynological approach will provide 1) a new insight on the dispersal of different TerrOM sources in river-dominated coastal zones and their possible interaction with marine productivity; 2) trends and spatial distribution of the marine OM produced in the marine realm.

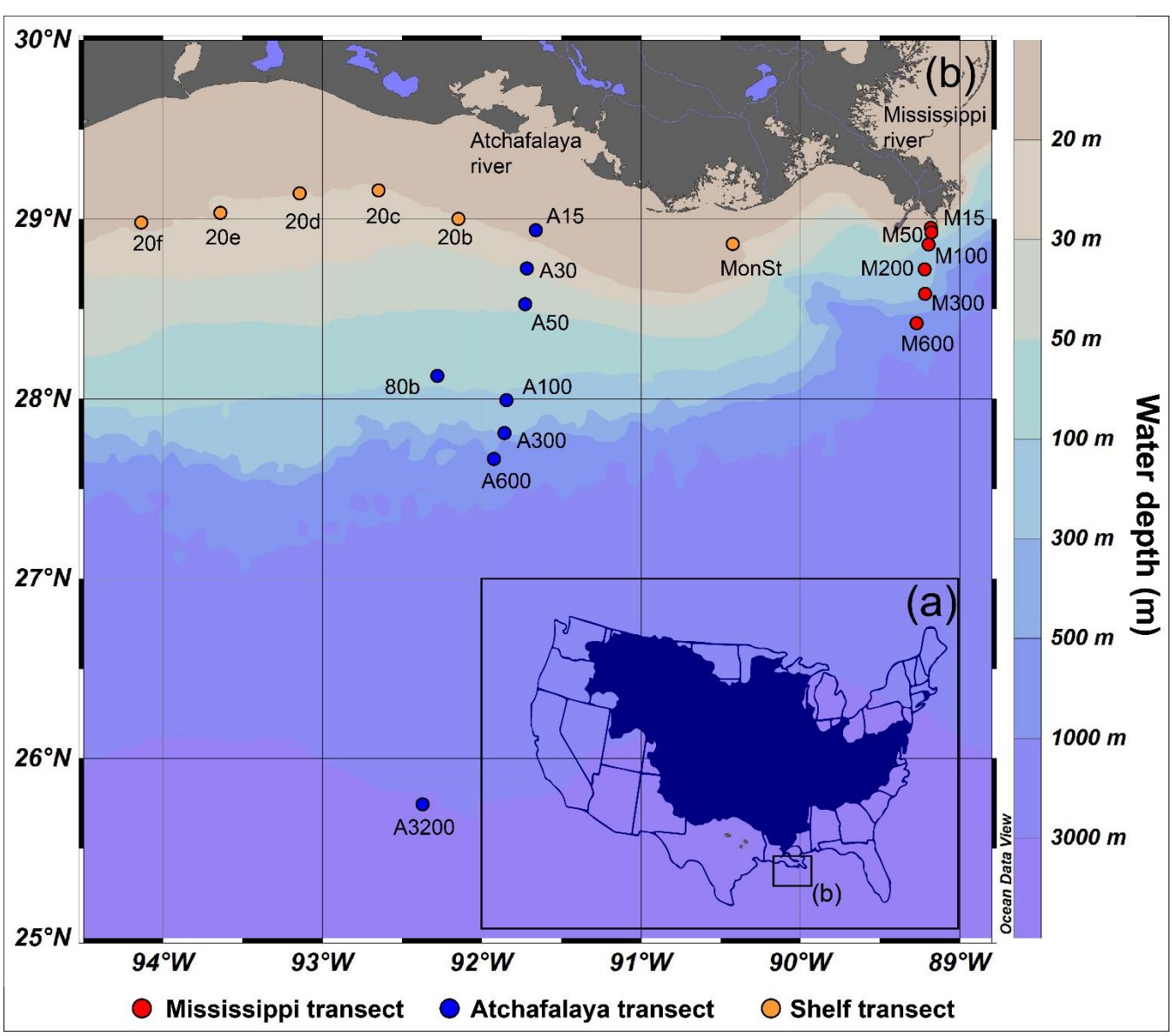

**Figure 1: a) The catchment of the Mississippi and Atchafalaya River System, and b) the locations of marine surface sediments in the northern GoM used for this study. Surface sediments have been collected along three transects; the land-sea transects of the Mississippi River (15-600 m water depth, samples M15, M50, M100, M200, M300 and M600) and the Atchafalaya River (15-3200 m water depth, samples A15, A30, A50, 80b, A100, A300, A600 and A3200) and a shelf transect following the river plume at the 20 m isobar (samples 20f, 20e, 20d, 20c, 20b, MonSt).**

## 2 Proxy background

### 2.1 Lipid biomarker proxies

To disentangle the composition of OM in the surface sediments of the northern GoM, we use concentrations and ratios of several lipid biomarkers. Glycerol dialkyl glycerol tetraethers (GDGTs) are membrane lipids of marine archaea and a group of largely unknown bacteria, which produce isoprenoid GDGTs (isoGDGTs; Koga et al., 1993), and branched GDGTs (brGDGTs; Sinninghe Damsté et al., 2000), respectively. Although the exact producer(s) of brGDGTs is still unknown, they were initially thought to be solely produced in soils and peats. Based on similarities in their concentration to that of

Acidobacteria in a peat profile (Weijers et al., 2009), and the detection of a specific brGDGT in an Acidobacterial culture (Sinninghe Damsté et al., 2011, 2018; Chen et al., 2022; Halamka et al., 2022), the phylum of Acidobacteria likely hosts their source organisms, although other bacterial phyla cannot be discarded (Weber et al., 2018; De Jonge et al., 2021). The Branched and Isoprenoid Tetraether (BIT) index (Hopmans et al., 2004), uses the presumed soil origin of brGDGTs relative to that of crenarchaeol, an isoGDGT that is exclusively produced by Thaumarchaeota (Sinninghe Damsté et al., 2002), to determine the

relative input of soil-derived OM in a marine environment. Accordingly, high BIT index indicates more soil input and a low BIT index a larger marine contribution to the sediment. However, recent studies have shown that brGDGTs can also be produced in rivers (e.g., Zell et al., 2013; De Jonge et al., 2014a), lakes (e.g., Weber et al., 2018; van Bree et al., 2020) and the coastal marine environment (e.g., Peterse et al., 2009; Zhu et al., 2011; Sinninghe Damsté, 2016), while also crenarchaeol can be produced in soils (Weijers et al., 2006), and thus the BIT index should be applied with care in the assessment of terrestrial

OM sources. Fluvial and marine in situ contributions to the brGDGT pool can be recognized by distinct traits in their molecular structure. For example, brGDGTs produced in rivers appear to be dominated by compounds that have a methylation at the C-6 position of their alkyl chain(s), as opposed to position C-5 that is more common for brGDGTs occurring in soils (De Jonge et al., 2014b; Warden et al., 2016; Kirkels et al., 2020). This difference can be quantified using the Isomer Ratio (IR), where a higher IR indicates a larger contribution of 6-methyl brGDGTs (De Jonge et al., 2014b). In addition, brGDGTs produced in

the coastal marine environment can be recognised by a higher number of cyclopentane rings present in the tetramethylated brGDGTs, which is captured in the #rings$_{tetra}$ ratio (Sinninghe Damsté, 2016). A value of >0.7 of the #rings$_{tetra}$ ratio is considered to represent a non-soil contribution to the signal, as these values are typically not reached in modern surface soils, but are common in alkaline shelf sediments of the ocean (Sinninghe Damsté, 2016).

We use the concentration of the isoGDGT crenarchaeol in the GoM sediments to constrain the ecological niche of

Thaumarchaeota, which are ammonia oxidizers and, therefore, play an important role in the nitrogen cycle (e.g., Wuchter et

al., 2006). As ammonia is an important breakdown product produced during the degradation of marine OM, the concentration of crenarchaeol can be used as a proxy for marine productivity.

Long-chain diols are biomarkers characterized by a long $n$-alkyl chain and two alcohol groups, attached at the C-1 and a mid-chain position. In marine settings, the diols that are commonly found are $C_{28}$ and $C_{30}$ 1,13-diols, the $C_{28}$ and $C_{30}$ 1,14-diols and
the $C_{30}$ and $C_{32}$ 1,15-diols. The 1,14-diols have been detected in *Proboscia* diatoms (Sinninghe Damsté et al., 2003). Also the 1,13- and 1,15- $C_{28}$ and $C_{30}$ diols are commonly found in marine settings, although their main source remains unclear (Rampen et al., 2012; Balzano et al., 2018). In contrast, $C_{32}$ 1,15-diols have been found in freshwater lakes and stagnant waters where they are produced by eustigmatophyte algae (Volkman et al., 1992; Shimokawara et al., 2010; Lattaud et al., 2021). Based on high fractional abundances of the $C_{32}$ 1,15-diol in coastal zones receiving significant river input (Rampen et al., 2014a; de Bar
et al., 2016), Lattaud et al. (2017) proposed that its relative abundance compared to the $C_{28}$ 1,13- and $C_{30}$ 1,13- and 1,15 diols (expressed in the $F_{C32\ 1,15}$ ratio) could be used as a proxy for fluvial organic matter input in coastal zones. In addition, we will use the concentration of the 1,14-diols as an indicator of diatom productivity.

Long-chain $n$-alkanes are derived from the epicuticular waxes of higher plants and widely used as markers for higher plants (Eglinton and Hamilton, 1963). These leaf waxes can be transported to the marine realm by fluvial and aeolian transport
(Gagosian et al., 1987). A strong odd-over-even carbon number predominance is an indication that the $n$-alkanes are derived from higher plants and is quantified in the Carbon Preference Index (CPI; Bray and Evans, 1961). The chain length of $n$-alkanes is presumed to be influenced by climate, where longer chain lengths are linked to warmer and drier conditions (Poynter et al., 1989). The Average Chain Length (ACL) is thus commonly used as indicator of vegetation type, where grasses produce $n$-alkanes with longer chain lengths than trees and shrubs (Bray and Evans, 1961). Here, we use the concentration of odd
carbon numbered $n$-alkanes derived from higher plants in a specific range ($C_{29}$-$C_{35}$) as a proxy for terrestrial plant input.

Higher plants furthermore produce sterols like campesterol, stigmasterol and β-sitosterol (Volkman, 1986; Meyers, 1997; Lütjohann, 2004), the presence of which has been used as a proxy for plant-derived OM (e.g., Waterson and Canuel, 2008; Xiao et al., 2013; Rontani et al., 2014). However, algal sources of these sterols have also been identified in marine algae (Rampen et al., 2010; Volkman, 2016). This hampers the use of these sterols as specific markers for higher plants and requires
comparison with other plant markers. Similarly, brassicasterol and dinosterol are often associated with diatoms and dinoflagellates, respectively, although other marine algal groups can also produce these sterols (Volkman, 2016; Sangiorgi et al., 2005; Rampen et al., 2010). We here use these sterols as additional proxies for marine OM and possibly higher plant OM, after careful comparison with other terrestrial and marine OM proxies.

Finally, we will use the presence and concentration of $n$-$C_{37}$ alkenones that are produced by haptophyte algae (Volkman et al.,
1980) as indicators for marine productivity by haptophytes.

## 2.2 Palynological proxies

Dinoflagellate cysts, pollen and spores were used as proxies for marine primary productivity and plant input, respectively. Dinoflagellates are protists that live in the upper water column and ~15% of the species produce organic-walled resting cysts

(dinocysts) during their life cycle (Head, 1996). Dinocysts can be preserved in sediments and their assemblages can be used as proxy for surface waters nutrient availability and productivity, upwelling, salinity, and temperature (Reid and Harland, 1977; Rochon et al., 1999; Zonneveld et al., 2013). Most dinoflagellate species are obligate autotrophs or heterotrophs, some are mixotrophs. Autotrophic species are commonly found in the photic zone, while heterotrophic species feed on organic debris and other organisms, including diatoms, bacteria, and other dinoflagellates, and therefore live in water with high primary productivity and/or organic matter input (e.g., Gaines and Taylor, 1984). For this reason, the percentage of cysts of heterotrophic dinoflagellates in assemblages is used to indicate primary productivity (e.g., Sangiorgi and Donders, 2004). It has been argued that the total dinocyst concentration (number dinocysts g$^{-1}$ sediment) in sediments is a better indicator of dinoflagellate productivity and total primary marine productivity (Zonneveld et al., 2009; Hardy et al., 2016) as cysts of heterotrophic and autotrophic dinoflagellates have different preservation potential (e.g., Zonneveld et al., 2010). Here, we use both the percentage of heterotrophic dinocysts (% Heterotrophs) in the assemblages and the total dinocyst concentration as indicator for marine productivity. To facilitate the comparison between dinocyst concentrations and (marine) biomarker concentrations, we normalised the dinocyst concentrations to g TOC (dinocysts g$^{-1}$ TOC). Routinely, dinocysts concentrations are presented as dinocysts g$^{-1}$ dry sediment.

Pollen grains and spores are produced by vegetation and occur widespread in coastal marine sediments. Most pollen are fluvially transported, resulting in high concentrations in coastal sediments in the proximity of river mouths (e.g., Heusser, 1988). Further offshore, pollen is usually less abundant, and wind-blown pollen (mainly *Pinus*) represent a relatively large contribution to the assemblage (Mudie and McCarthy, 1994; Chmura et al., 1999; Donders et al., 2018). To exclude aeolian transport, *Pinus* pollen was not included in the total pollen concentration. This pollen concentration, normalised to g TOC (pollen g$^{-1}$ TOC), is here used as a proxy for terrestrial plant input transported by rivers.

## 3 Materials and Methods

### 3.1 Study site

With a catchment of 3.3*10$^6$ km$^2$ (Milliman and Syvitski, 1992), the MAR is one of the largest river basins worldwide, and the largest watershed in North America (Fig. 1a). It is responsible for the input of over 90% of freshwater, nutrients, and suspended material that accumulate in the northern GoM (Deegan et al., 1986; Rabalais et al., 2007). The Mississippi River (MR) has a mean annual water discharge of ~18,400 m$^3$ s$^{-1}$, while the Atchafalaya River (AR) discharges ~4,400 m$^3$ s$^{-1}$ (Bianchi and Allison, 2009; data from USGS). In the 1960s, a control structure was completed in the river, which diverted a combined 30% of the stream flow of the Lower MR and Red River into the AR (Reuss, 2004). Upon discharge, the MAR water flow is directed westward along the Louisiana shelf under the influence of local wind stress. Annual wind patterns in the GoM area are predominantly westward, and wind comes directly from land during winter months (Zavala-Hidalgo et al., 2014). Subsequently, the MAR water flow is predominantly directed westward along the Louisiana shelf upon discharge. Further offshore, the surface circulation is mainly controlled by the Loop Current, which in summer brings oligotrophic Caribbean

surface waters to the northern GoM (Sturges and Evans, 1983; Schmitz Jr et al., 2005). When the Loop Current is well extended, it can interact with the shelf break (100 m isobar) east of the MR mouth, and episodically west of the MR, although this mainly happens with unusual conditions via detached warm eddies (Schiller et al., 2011).

Annual particle loads of the MR and AR reach 115 and 57 Pg yr$^{-1}$, respectively. These high loads result in sedimentation rates of >10 cm yr$^{-1}$ close to the mouths of both rivers (Bianchi et al., 2002; Santschi and Rowe, 2008). The shelf areas of the MAR differ in morphology, which influences the sediment dispersal; the high discharge of the MR has led to rapid delta progradation, resulting in a steep shelf where most of the suspended particles is transported offshore (Corbett et al., 2006), whereas the AR discharges mainly into the Atchafalaya Bay (Neill and Allison, 2005; Hetland and DiMarco, 2008; Xu et al., 2011), which has resulted in a more gently sloped part of the Louisiana shelf. The MR mouth is located close to the Mississippi canyon, the formation of which is generally linked to channel entrenchment of the MR during sea level low stands (Coleman et al., 1982). Subsequently, a portion of the sediments discharged by the MR is transported downslope into the Mississippi canyon (Coleman, 1988; Bianchi et al., 2006; Sampere et al., 2008; 2011), resulting in higher sediment export rates compared to the Atchafalaya River (McKee et al., 2004). These differences in morphology also influence physical shelf processes such as water column stratification; the steeper morphology of the MR allows for a well-defined pycnocline, whereas the shallower shelf (<20 m) close to the AR mouth prevents the development of a stratified water column (Hetland and DiMarco, 2008). This water column stratification plays a large role in the formation of hypoxia, by restricting the regeneration of oxygen towards the bottom waters.

Highest freshwater discharge generally occurs in early spring, while the peak in nutrient load delivered to the northern GoM happens around June (Rabalais et al., 2007). The combined effect of algal blooms triggered by the high nutrient input and water column stratification in summer months, where waters are deep enough to form a well-defined pycnocline, causes the formation of a seasonal hypoxic zone (O$_2$ concentrations of <2 mg l$^{-1}$) that currently covers 23,000 km$^2$ at the seafloor (Lohrenz et al., 1990; 1997; Rabalais and Turner, 2019). These lower oxygen conditions are most common between 10 and 30 m water depth on the shallow continental shelf in the northern GoM. In the last decades, hypoxic conditions became more widespread due to excess nutrient input from the MR derived from agricultural activities (e.g. Rabalais et al., 2002). Hypoxia can be further enhanced by respiration of organic matter in the water column or in shelf sediments, which further consumes oxygen (Hetland and DiMarco, 2008; Bianchi et al., 2010).

### 3.2 Sample collection

Marine surface sediments (0-2 cm) were collected during a research cruise with the R/V *Pelagia* in February 2020. Multicores were acquired with an Oktopus multicoring apparatus at 20 locations, covering a transect along the 20 m isobar on the shelf in the direction of the river plume and land-sea transects from the MR and AR at depths ranging from 15-600 m and 15-3200 m, respectively (Fig. 1b). During sampling, CTD (conductivity, temperature and density) water column profiles and oxygen measurements indicated that conditions were oxic at all sample locations. At most sites, the sediment were muddy, except for

the western shelf, where the sediments had a sandier nature. Sediments were stored at -20 °C on board and remained frozen during transport to the laboratory. All sediments were freeze-dried prior to further analysis.

Part of our sampling sites are located within a seasonal hypoxic zone, and experiences lower oxygen levels in bottom waters in summer (Rabalais et al., 2002). The availability of oxygen represents a large factor in the preservation potential of TerrOM (Hedges et al., 1999) and can therefore influence the dispersal patterns of TerrOM (Bianchi et al., 2010). During hypoxic conditions, high riverine discharge can facilitate the rapid burial of OM. Nevertheless, sediment accumulation rates found on the Louisiana shelf ranged from ~1.5 cm yr$^{-1}$ on the shallow shelf, towards 0.2-0.3 cm yr$^{-1}$ on the slope (Lenstra et al., 2022).

Therefore, the upper 0-2 cm sediment analysed, which represents at least 1 year deposition time, integrates multiple years of seasonally varying oxygen conditions. Still, sediment accumulation rates can vary substantially on a seasonal scale, depending on river discharge, biological processes, and hydrological factors like waves and tides (McKee et al., 2004). For example, sediment accumulation rates near the MR delta derived from short-lived radionuclides ([7]Be and [234]Th) have shown annual and seasonal variations that are larger than the average decadal sedimentation rates calculated with [210]Pb (Corbett et al., 2004),

suggesting that active sediment reworking and possible export of particles off- and along-shore takes place. Nevertheless, albeit not completely correct, we here assume that the integrated, multiple year signal in our surface sediments reduces any OM burial and preservation biases among locations in our set of GoM sediments.

### 3.3 Bulk sediment analysis

Freeze-dried and homogenized sediments (ca. 0.3 g) were treated with 1 mL HCl overnight to remove inorganic carbon. The TOC, TN and $\delta^{13}C_{org}$ were analysed using an Elemental Analyzer (Fisons Instruments NA 1500) coupled to an isotope ratio mass spectrometer (IRMS, FinniganMat Delta Plus). TOC and TN are expressed as weight percentage (wt.%) of the dried sediment and have an error of ± 0.01 wt.% (TOC) and ± 0.001 wt.% (TN) based on the standard deviation of replicate runs of lab standards nicotinamide and IVA. These values were used to derive the molar C/N ratios. The $\delta^{13}C_{org}$ values are reported

relative to the Vienna Pee Dee Belemnite standard (VPDB). Reproducibility of $\delta^{13}C_{org}$ measurements was usually better than 0.04‰ based on lab standards.

### 3.4 Lipid biomarker analysis

Lipid biomarkers were extracted from ~1 g of freeze dried and homogenized sediment with 25 mL of dichloromethane (DCM):methanol (9:1, v/v), using the microwave extraction system Milestone Ethos X (MEX). The total lipid extract was

255 passed over a small sodium sulphate column to remove any remaining water prior to separation into an apolar, neutral and polar fraction, by passing it over a small column with activated aluminium oxide and solvent mixtures of hexane:DCM (9:1), hexane:DCM (1:1) and DCM:methanol (1:1) as the respective eluents.

The apolar and neutral fractions, containing *n*-alkanes and alkenones, respectively, were dissolved in 10 μL hexane and co-injected with an external squalene standard on-column on a gas chromatograph coupled to a flame ionisation detector (GC-

FID, Hewlett Packard 6890 series). The GC-FID was operated with helium as carrier gas at a constant pressure of 100 kPa, and an oven temperature starting at 70 °C and rising first to 130 °C at 20 °C min$^{-1}$ and then to 320 °C at 4 °C min$^{-1}$, at which it was held for 10 min. Concentrations of *n*-alkanes and alkenones were determined by relating their peak areas with that of the squalene standard. The CPI and ACL were calculated using the equations from Marzi et al. (1993) and Poynter et al. (1989), respectively:

$$CPI = ((C_{23} + C_{25} + C_{27} + C_{29} + C_{31} + C_{33}) + (C_{25} + C_{27} + C_{29} + C_{31} + C_{33} + C_{35})) / (2 * (C_{24} + C_{26} + C_{28} + C_{30} + C_{32} + C_{34})) \tag{1}$$

$$ACL = (25C_{25} + 27C_{27} + 29C_{29} + 31C_{31} + 33C_{33} + 35C_{35})/(C_{25} + C_{27} + C_{29} + C_{31} + C_{33} + C_{35}) \tag{2}$$

A known amount of a synthetic $C_{46}$ glycerol trialkyl glycerol tetraether (GTGT) standard was added to the polar fraction (Huguet et al., 2006), which was subsequently dissolved in 1.5 mL hexane:isopropanol (99:1, v/v) and passed over a 45 μm polytetrafluorethylene (PTFE) filter. Analysis of GDGTs was done using high performance liquid chromatography/mass spectrometry (HPLC/MS) using the method of Hopmans et al. (2016). Peaks were detected via selected ion monitoring (SIM), using m/z 744 for the internal standard, m/z 1302, 1300, 1298, 1296 and 1292 for isoGDGTs and m/z 1050, 1036, 1034, 1032, 1022, 1020 and 1018 for brGDGTs. For GDGT proxy calculations, the roman numerals refer to the concentrations of the GDGTs as listed in Sinninghe Damsté, (2016). The BIT index was calculated according to Hopmans et al. (2004) and includes the 5- and 6-methyl brGDGTs:

$$BIT = [(Ia) + (IIa) + (IIIa) + (IIa') + (IIIa')]/[(Ia) + (IIa) + (IIIa) + (IIa') + (IIIa') + (Cren)] \tag{3}$$

The isomer ratio (IR) expresses the fractional abundance of 6-methyl brGDGTs over 5- and 6-methyl brGDGTs (De Jonge et al., 2014b):

$$IR = [(IIa') + (IIb') + (IIc') + (IIIa') + (IIIb') + (IIIc')]/[(IIa') + (IIb') + (IIc') + (IIIa') + (IIIb') + (IIIc') + (IIa) + (IIb) + (IIc) + (IIIa) + (IIIb) + (IIIc)] \tag{4}$$

The number of rings of tetramethylated brGDGTs (#rings$_{tetra}$) was calculated using the equation of Sinninghe Damsté (2016):

$$\#rings_{tetra} = [(Ib) + 2*(Ic)/[(Ia) + (Ib) + (Ic)] \tag{5}$$

Diols and sterols were analysed by silylation of an aliquot of the polar fraction, using 10 μL pyridine and 10 μL N$_2$O-bis(trimethylsilyl)trifluoroacetamide (BSTFA) and heating to 60 °C for 20 min. The silylated fractions were dissolved in 30 μL ethyl acetate and co-injected with the squalene standard on-column on the GC-FID. For peak identification and qualitative integration, each sample was also analysed using GC- mass spectrometry (Thermo Trace Ultra GC connected to Finnigan Trace DSQ mass spectrometry, GC-MS DSQ), with a mass range *m/z* 50-800, using helium as carrier gas with a constant flow rate of 2.0 ml/min. To ensure analytical robustness, all instruments were regularly checked using lab standards. Diols that were considered for analysis were $C_{28}$ 1,14 (m/z 299), $C_{28}$ 1,13, $C_{30}$ 1,15 (m/z 313), $C_{30}$ 1,14 (m/z 327), $C_{30}$ 1,13 and $C_{32}$ 1,15 (m/z 341). The $F_{C32\ 1,15}$ is calculated according to Lattaud et al. (2017) and uses the areas of the 1,13-and 1,15-diols:

$$F_{C32\ 1,15} = (AC_{32}\ 1,15 / (AC_{32}\ 1,15 + AC_{30}\ 1,15 + AC_{28}\ 1,13 + AC_{30}\ 1,13)) * 100 \qquad (6)$$

Concentrations for all biomarkers have been normalized to TOC.

### 3.5 Palynological processing

Freeze dried and homogenized sediments (ca. 20 g) were selected for palynological analysis. A *Lycopodium clavatum* tablet containing 19,855 spores (± 2.62%) was added to each sample to enable dinocysts, and pollen and spores quantification (Stockmarr, 1971; Wood, 1996). Sediments were treated with 30% HCl and 40% cold HF at room temperature to remove carbonates and silicates, respectively. No oxidation was performed. After acid treatment, the sediments were sieved using a fine mesh and an ultrasonic bath to isolate the 10-250 μm fraction. Subsequently, residues were made and mounted on microscope slides with glycerine jelly. Between 116 and 245 dinocysts (median 203) and 186 – 300 pollen grains (median 258) were counted using optical microscopy at 400x magnification. The dinocyst taxonomy follows Williams et al. (2017) and dinocysts were identified following Rochon et al. (1999) and Zonneveld and Pospelova (2015). Pollen and spores identification was based on Willard et al. (2004).

### 3.7 Data processing

Isosurface plots were constructed using Ocean Data View software (Schlitzer, 2015), using weighted average gridding. Principle Component Analysis (PCA) was performed on relative concentrations of biomarkers, pollen, and dinoflagellate cysts using Past version 4.04 (Hammer et al., 2001), where all biomarker, pollen and dinoflagellate cyst concentrations were normalized against their maximum value (min-max normalization) to facilitate comparison between all proxies.

### 4 Results

### 4.1 Bulk parameters

The total organic carbon (TOC) content (in wt.%) of the surface sediments varied between 0.05 and 1.63 wt.% and was much lower (≤0.65%) on the western portion of the shelf than close to the MAR (>1.42%) or offshore (Fig. 2a). The molar C/N ratios varied between 8.2 and 11.9. The highest C/N ratios were found close to the MAR, while C/N values decreased further from shore (Fig. 2b). The $\delta^{13}C_{org}$ values ranged from -21.8 to -24.2‰, with lower $\delta^{13}C_{org}$ values along the shore and higher values further away from the MAR (Fig. 2c).

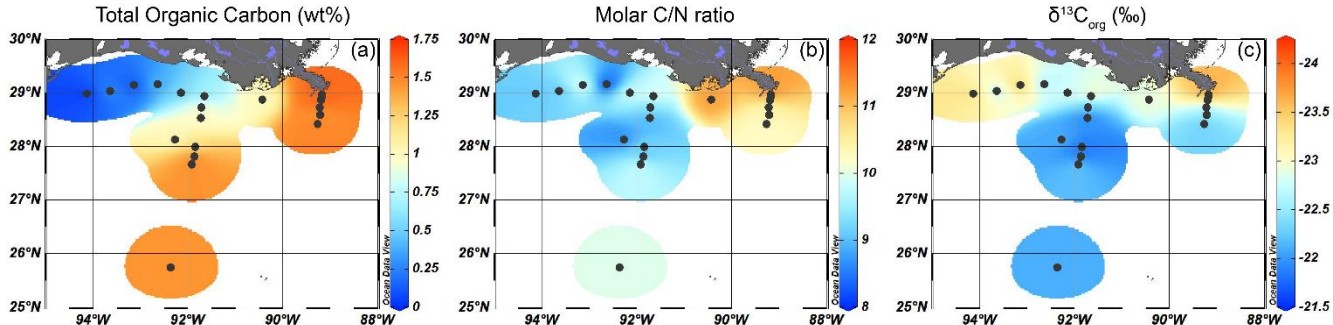

**Figure 2: Isosurface plots for bulk sediment properties for surface sediments in the GoM for a) TOC content (wt.%), b) the molar C/N ratio, c) $\delta^{13}C_{org}$ (‰).**

## 4.2 Lipid biomarkers

BrGDGTs were detected at all sites, with concentrations of 40-70 μg g⁻¹ TOC close to the MR, decreasing to <20 μg g⁻¹ TOC on the shelf, and then to <10 μg g⁻¹ TOC in the deeper ocean (Fig. 3a). Crenarchaeol was the most dominant isoGDGT (mean 65 ± 5% of all isoGDGTs) with absolute concentrations varying between 30 and 400 μg g⁻¹ TOC. Concentrations of crenarchaeol were lowest close to the MR, and then increased towards the shelf. In the surface sediments in deeper waters (> 100 m), crenarchaeol concentrations were always >90 μg g⁻¹ TOC (Fig. 4a). These concentrations translate into BIT index

values ranging from 0.14-0.57 close to the MR with lower values (<0.10) towards open sea (Fig. 3b).

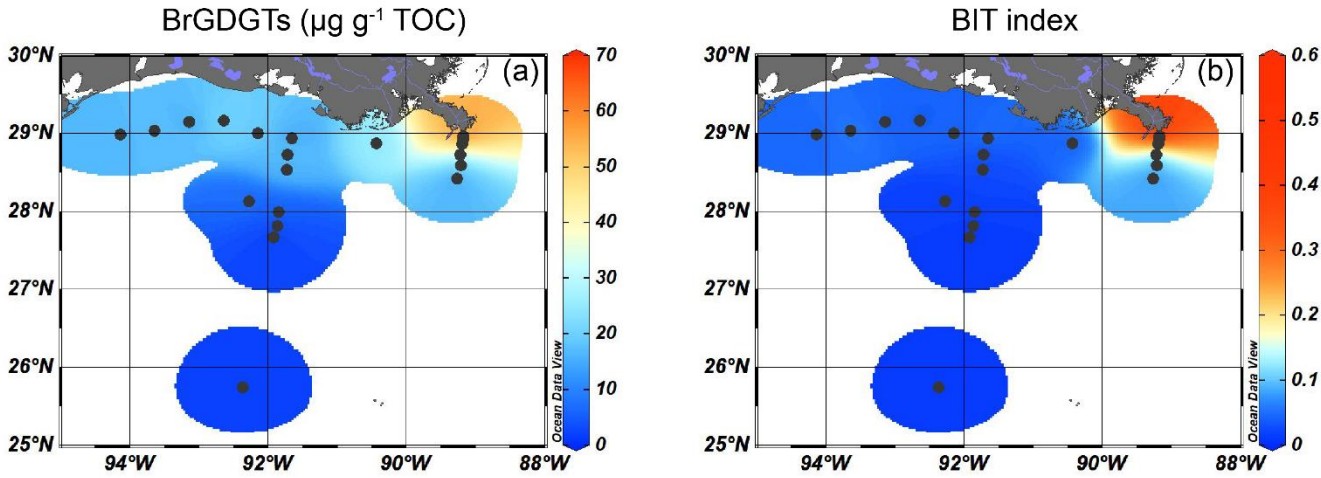

**Figure 3: Isosurface plots for proxies for SMOM input for surface sediments in the GoM: a) the concentration of brGDGTs (μg g⁻¹ TOC), and b) the BIT index.**

Long-chain diols were detected at 19 of the 20 sites, with concentrations ranging from 7 – 70 μg g⁻¹ TOC. At the westernmost shelf location, where the TOC concentration was the lowest (0.05 wt.%), diols were not detected. The highest concentration of 1,14-diols was found on the shelf and on the shallow part (<100 m water depth) of the Atchafalaya transect (12-18 μg g⁻¹

TOC), with lower concentrations on the Mississippi transect and in deeper waters (<8 µg g⁻¹ TOC, Fig. 4b). The absolute abundance of the $C_{32}$ 1,15-diol varied from 1 µg g⁻¹ TOC in the deepest waters to >10 µg g⁻¹ TOC on the shelf close to the AR

(Fig. 5a), while the $F_{C32\ 1,15}$ ranged from 6-51% and showed a trend with higher values close to the MR mouth, and lower values further away (Fig. 5b).

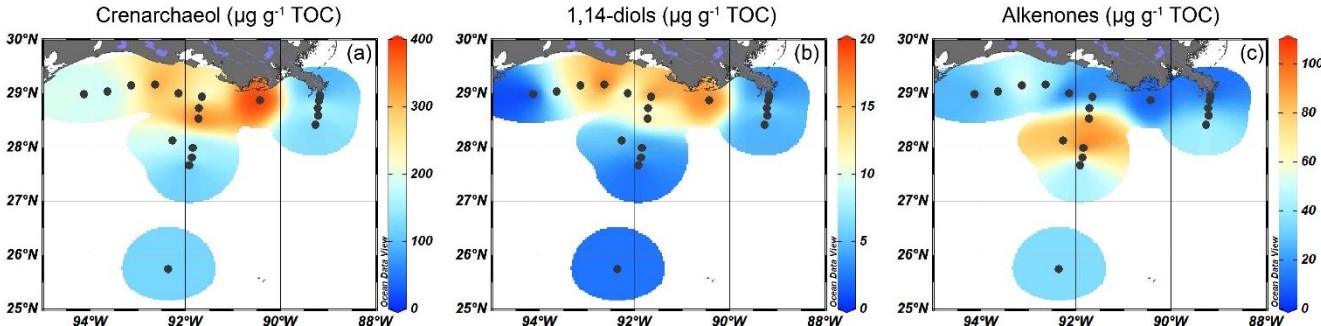

Figure 4: Marine OM production in the GoM, as indicated by isosurface plots of TOC-normalized concentrations of a) crenarchaeol
produced by Thaumarchaeota, b) long-chain 1,14-diols produced by *Proboscia* diatoms and c) alkenones produced by haptophyte algae,

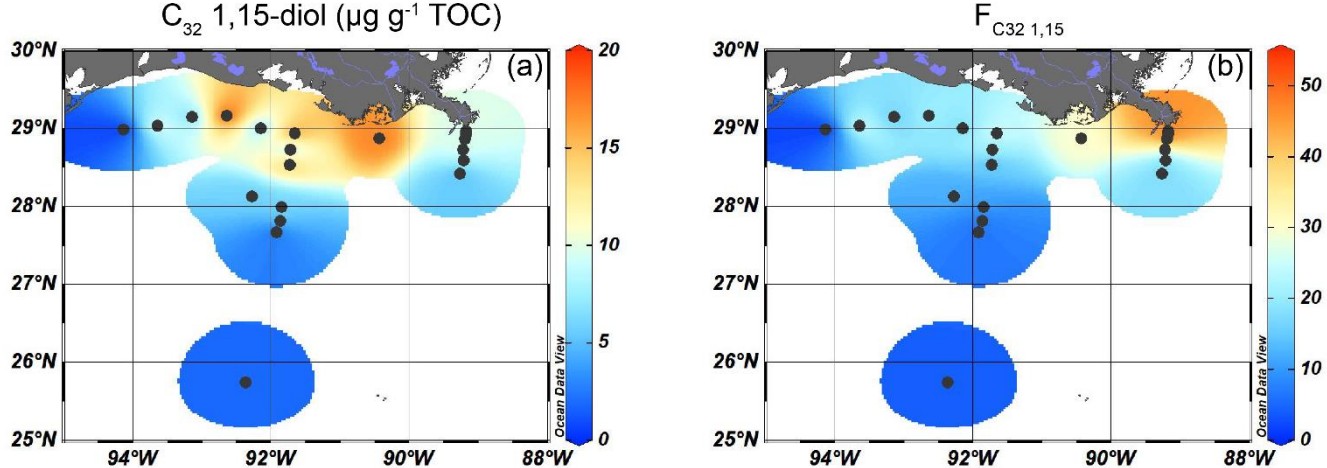

Figure 5: Isosurface plots of proxies for fluvially-discharged fresh water TerrOM input for surface sediments in the GoM: a) the
concentration of $C_{32}$ 1,15-diols (µg g⁻¹ TOC) and b) the $F_{C32\ 1,15}$ index.

Alkenones were mostly found in sediments from a water depth between 50 and 300 m in the Atchafalaya transect (70 – 110 µg g⁻¹ TOC), while their abundance in sediments from the shelf and from deeper waters was much lower (3 – 60 µg/ g⁻¹ TOC) (Fig. 4c).

Five sterols were identified at all sites. Total sterol concentrations ranged from 25 µg g⁻¹ TOC in deeper waters, to >400 µg g⁻¹ TOC on the shelf (Fig. 6a). The highest total sterol concentrations (677 µg g⁻¹ TOC) were found in between the MR and AR

deltas. Due to the presumed mixed terrestrial plant and algal sources of these individual sterols, their relative abundance was also plotted to identify spatial differences in their occurrence (Fig. 6b-f). β-sitosterol was the most abundant, accounting for ~30% of the sterol assemblage on the shelf and up to 48% in deeper waters, where total sterol concentrations were low (Fig. 6b). Dinosterol was most abundant at the deeper MR and AR transects (>25% of total sterols), while brassicasterol was most abundant on the shelf and close to the MR (>20%) (Fig. 6c, d). The relative abundance of stigmasterol was highest in sediments from ca. 80 m water depth of the Atchafalaya transect (20%), while that of campesterol was high close to the MR (13%), Fig. 6e,f.

Long-chain $n$-alkane ($C_{29}$-$C_{35}$) concentrations varied between 23 and 210 µg g$^{-1}$ TOC. They were most abundant near the mouth of the MAR but were also consistently present on the shelf and more seaward, albeit in increasingly lower concentrations (from 70-130 to ~50 µg g$^{-1}$ TOC, Fig. 7a). At 19 of 20 sites, the $n$-alkanes showed a clear odd-over-even preference, reflected by a CPI ranging from 2.3-6.8, pointing to a predominant higher plant origin. The only exception was again the most western shelf site, where the CPI was 0.8. The ACL slightly varied between 29.4-30.4, without a spatial trend.


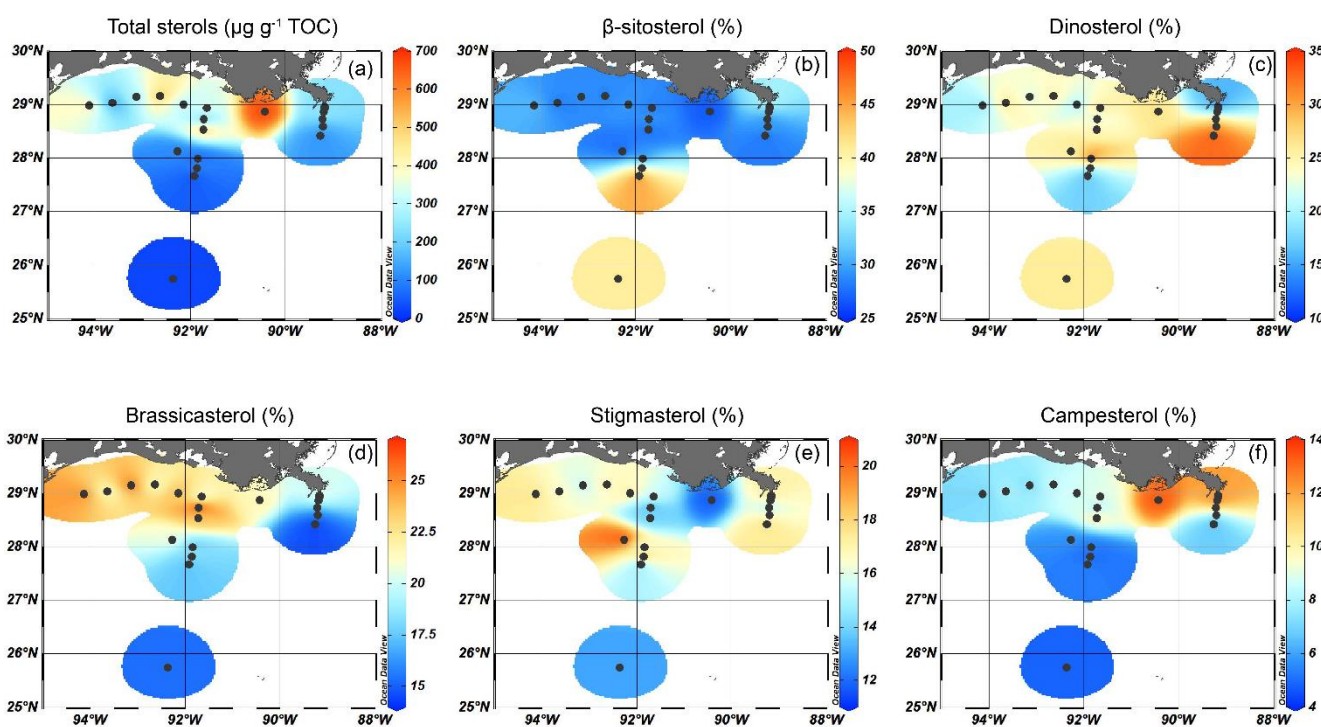

**Fig. 6: Isosurface plots showing a) the total sterol concentration and b-f) relative abundances of the individual sterols; b) β-sitosterol, c) dinosterol, d) brassicasterol, e) stigmasterol, f) campesterol.**

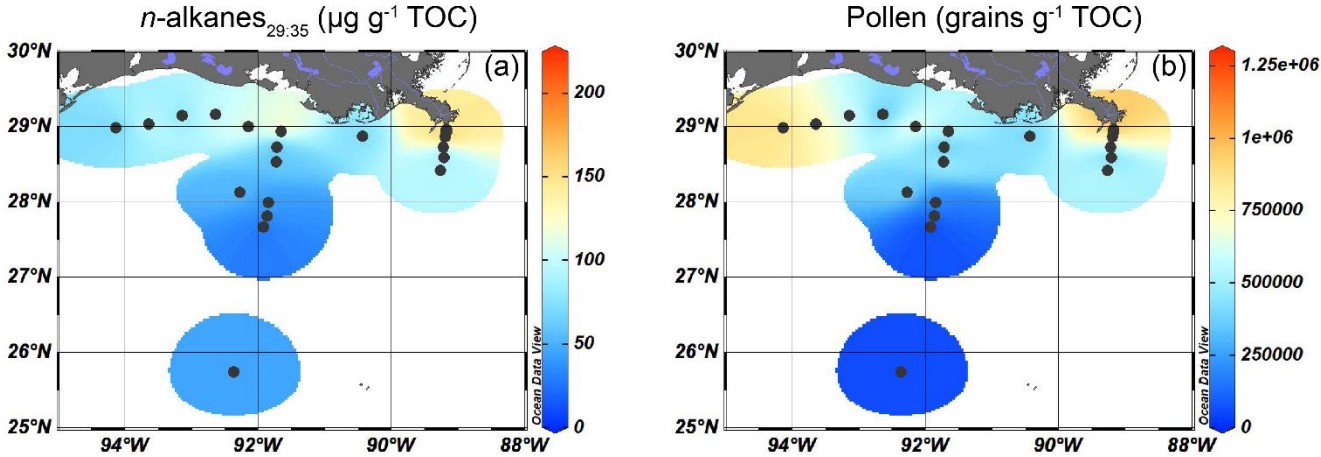

**Figure 7: Isosurface plots for the input of higher plant-derived OM in surface sediments of the GoM, based on OC-normalized concentrations of a) long-chain (C$_{29}$-C$_{35}$) *n*-alkanes and b) pollen grains (excluding *Pinus* pollen).**

## 4.3 Palynology

All sediments analysed contained well-preserved marine (dinocysts) and terrestrial (pollen, spores) palynomorphs. The total dinocyst concentration was highest on the shelf and shallow Atchafalaya transect (<100 m, ~5.0*10$^5$ cysts g$^{-1}$ TOC), and lowest along the Mississippi transect (~ 1.0*10$^5$ cysts g$^{-1}$ TOC) and the deeper part of the Atchafalaya transect (>100 m, ~0.4 – 1.1*10$^5$ cysts g$^{-1}$ TOC, Fig. 8). In sediments close to the shelf and the MR, heterotrophic taxa dominated the assemblages (60-85% of total cysts), while autotrophic taxa were relatively more abundant in sediments from the open ocean (69-80%, Fig. 8). Pollen and spores were most abundant close to the MR mouth (1.2 10$^6$ pollen g$^{-1}$ TOC) and on the westernmost part of the shelf (8.8 10$^5$ pollen g$^{-1}$ TOC, Fig. 7b). Their concentration decreased with increasing distance from the coast and reached a minimum of 3.9 10$^4$ pollen g$^{-1}$ TOC at the deeper Atchafalaya transect.

**Figure 8: Isosurface plots of the TOC-normalized dinocyst concentration in GoM surface sediments overlain by pie-charts showing the relative contribution of autotroph and heterotroph species.**

### 4.4 Statistical analysis on biomarker and palynology concentrations

The PCA of normalized biomarker, pollen, and dinoflagellate cyst concentrations in the surface sediments showed a variance of 55.6% on PC1 and 19.7% on PC2 (Fig. 9). Almost all variables plot positively on PC1, together with shallow shelf (<20 m water depth) sediments. The only exception is the concentration of alkenones, which plot negatively on PC1, together with samples taken at >80 m water depth along the AR transect.. PC2 mostly separates terrestrial and marine markers; the concentrations of n-alkanes, pollen and brGDGTs that are all considered to reflect a higher plant or SMOM source plot positively, while the concentrations of marine biomarkers alkenones, crenarchaeol, and 1,14-diols plot negatively. The sterols and the $C_{32}$ 1,15-diols have generally low scores on this PC. Sediments from the MR transect scored positively on PC2, while the AR transect scored negative on PC2. Sediments from the shelf transect have generally low scores on this PC.

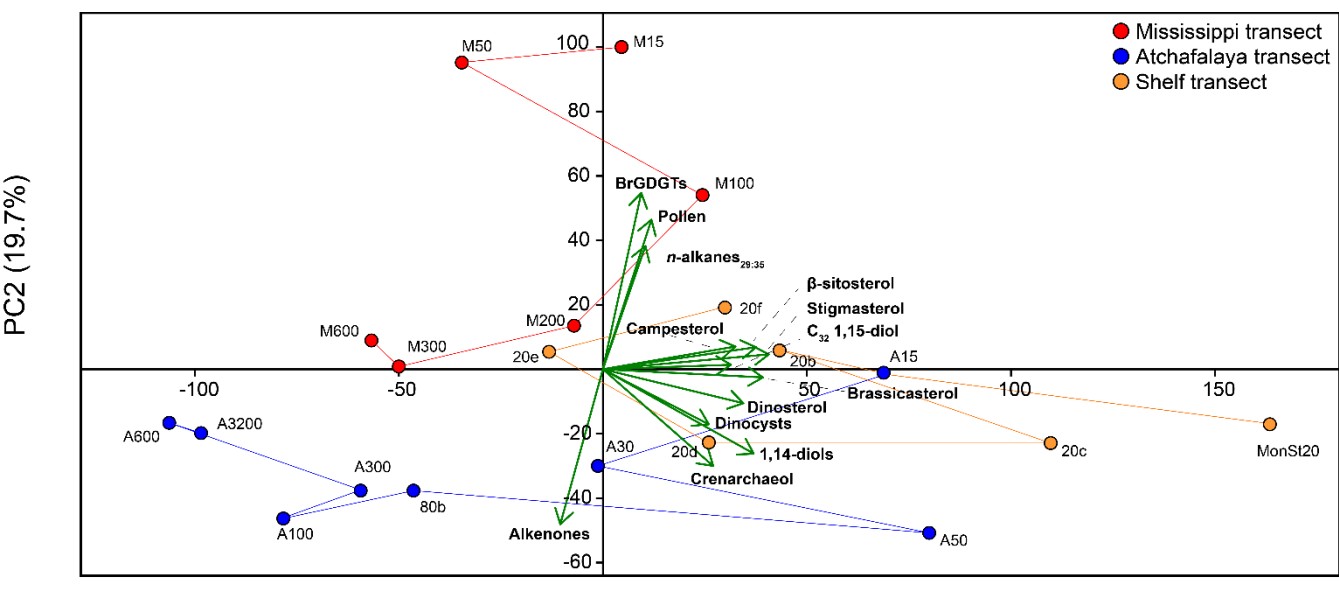

**Figure 9: PCA on concentrations of brGDGTs, pollen, *n*-alkanes29:35, the C32 1,15-diol, campesterol, β-sitosterol, stigmasterol, , brassicasterol, dinosterol, dinocysts, 1,14-diols, crenarchaeol and alkenones in surface sediments of the GoM. Lines connecting sample locations represent the scores of the different transects.**

## 5 Discussion

### 5.1 Spatial patterns of bulk OM parameters in surface sediments

The TOC content ranges from 0.7-1.6 wt.% on the MR and AR transects (Fig. 2a) but is substantially lower (0.05-0.53 wt.%) on the shelf transect, likely due to their sandy nature. In the studied surface sediments, the C/N ratio ranges from 8 to 12 (Fig. 2b), suggesting that the OM has a primarily marine origin, as the C/N ratio of marine OM typically ranges between 6-7, while the C/N ratios of terrestrial plant OM is >20 (Hedges et al., 1997). Close to the MR, the C/N ratio values are highest, and similar to the molar C/N ratio of 12.2 reported earlier for the MR delta (Bianchi et al., 2002), as well as those of 9-12 for the shelf (Waterson and Canuel, 2008; Liu and Xue, 2020).

The $\delta^{13}C_{org}$ values for sediments in the northern GoM are lower close to the MAR and on the shelf (<-22.6‰, Fig. 2c) than further offshore, where $\delta^{13}C_{org}$ is >-22.3‰. While the $\delta^{13}C_{org}$ signature of C3 higher plants, SMOM and freshwater algae generally ranges from ~-27‰ – -30‰, C4 plants and marine algae have $\delta^{13}C_{org}$ values of ~-15‰ and ~-21‰, respectively (O'Leary, 1981; Hedges et al., 1997). The $\delta^{13}C_{org}$ values in the GoM thus show a predominant marine signal, especially further offshore. However, contributions of TerrOM such as C4 plants could also be incorporated in this signal. For example, the distribution of lignin phenols in sediments of the Gulf of Mexico (GoM) showed that C4 plant material was transported further offshore and contributed to the OM-pool in areas with enriched $\delta^{13}C_{org}$ signatures (Goñi et al., 1997, 1998; Bianchi et al.,

2002). If $\delta^{13}C_{org}$ values alone would be used, this would have conventionally been interpreted as indicating a predominant marine origin of the OM. Therefore, we here employed lipid biomarkers, pollen and dinocysts to further disentangle the sources of OM in the surface sediments.

## 5.2 Sources and dispersal of fluvially discharged TerrOM

### 5.2.1 OM produced in freshwater

The input of fluvially produced OM into the northern GoM can be traced using the fractional abundance of the $C_{32}$ 1,15-diol ($F_{C32\ 1,15}$), which is predominantly produced in freshwater, relative to that of the other 1,13- and 1,15 long-chain diols that have a marine origin (Lattaud et al., 2017, 2021). The $F_{C32\ 1,15}$ is highest proximal to the MR (50%) and decreases to <5% further offshore (Fig 5b). To assess whether the trend in $F_{C32\ 1,15}$ is driven by a marine overprint or a progressive loss of the $C_{32}$ 1,15-diol upon river discharge, the trend in the sedimentary concentration of $C_{32}$ 1,15-diol was evaluated. Interestingly, the highest TOC-normalized concentration of the $C_{32}$ 1,15-diol is observed in sediments on the shelf west of the MR mouth, where $F_{C32\ 1,15}$ is low (Fig. 5). A possible explanation for this could be a westward transport of the $C_{32}$ 1,15-diols combined with a substantial marine production of $C_{28}$ and $C_{30}$ 1,13- and 1,15-diols that overprint the $C_{32}$ 1,15-diol concentrations, generating the low $F_{C32\ 1,15}$. In the PCA, the $C_{32}$ 1,15-diols score in between terrestrial (brGDGTs, *n*-alkanes) and marine (alkenones, crenarchaeol) proxies on PC2 (Fig. 9), suggesting a mixed origin of fluvial and marine sources. However, previous studies have shown that marine production of the $C_{32}$ 1,15-diol occurs in much smaller relative abundances than of the other long-chain diols and that the $C_{32}$ 1,15-diol is predominantly produced in freshwater or brackish environments (Rampen et al., 2014b; Lattaud et al., 2017). Moreover, hydrogen isotopes of the $C_{32}$ 1,15-diols from surface sediments in the northern GoM are more depleted close to the MAR, showing additional evidence of a freshwater source of the $C_{32}$ 1,15-diols (Lattaud et al., 2019). Thus, the spatial pattern the $C_{32}$ 1,15-diols with high abundances on the shelf and their position in the PCA plot suggests that that they are not immediately lost upon discharge but are transported with the river plume along the shelf upon discharge. However, a marine contribution cannot be excluded based on the PCA results.

### 5.2.2 Soil-microbial OM

An indication of the relative input of fluvially discharged SMOM may be provided by the BIT index (Hopmans et al., 2004). Only the two sites closest to the MR mouth have BIT index values >0.3, while the BIT index is always ≤0.1 for all other sites (Fig. 3b). The interpretation of the BIT index as a proxy for terrestrial OM input requires caution, as brGDGTs may also be produced in the coastal sediments and/or water column (e.g., Peterse et al., 2009; Xie et al., 2014; Sinninghe Damsté, 2016) and in rivers and lakes (Zell et al., 2013; De Jonge et al., 2014a; Weber et al., 2018). To assess the source(s) of brGDGTs in the northern GoM, the fractional abundances of tetra-, penta- and hexamethylated brGDGTs were plotted on a ternary diagram and compared to the position of global soils and peats (Fig. 10a; Dearing Crampton-Flood et al., 2019). A contribution of in situ produced marine brGDGTs can be recognized by an offset from the global soils (Sinninghe Damsté, 2016). The sediments

collected close to the MR and along the shelf transect plot within the cloud of datapoints formed by the soils from the global dataset. This supports the idea that the sediments characterized by the highest BIT index values (>0.3) receive a substantial brGDGT influx from soil erosion. On the other hand, sediments collected along the AR transect plot away from the cloud of data points formed by the global soil dataset. This is a clear indication that the brGDGTs at these sites are not exclusively soil-derived. Indeed, brGDGTs in sediments that are offset from the global soils are also characterized by a #rings$_{tetra}$ of 0.37-0.91

(mean of 0.59 ± 0.17 SD, Fig 10b), which is significantly higher than the #rings$_{tetra}$ reported for global soils (0.21 ± 0.20 SD) and soils from the Mississippi catchment (0.23 ± 0.20 SD, n=14; De Jonge et al., 2014a), clearly indicating contributions of sedimentary in situ production (cf. Sinninghe Damsté, 2016). High #rings$_{tetra}$ of >0.7 on the western shelf and on the Atchafalaya transect in the zone between 30-300 m water depth indicates that most production of brGDGTs takes place in situ (Fig. 10b). This matches the findings from other shelf areas, where in situ brGDGT production is most pronounced at 50-300

455 m water depth (e.g. Sinninghe Damsté, 2016; Ceccopieri et al., 2019).

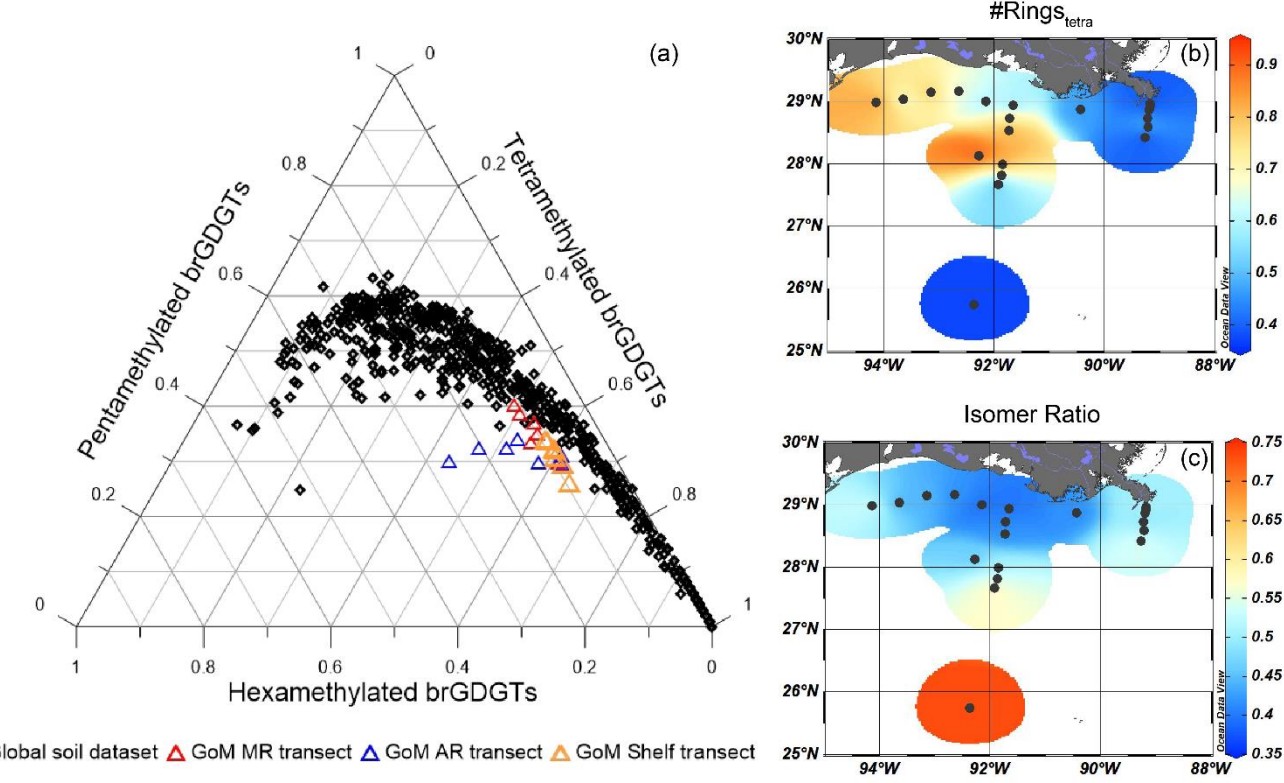

**Figure 10: Assessment of the sources of brGDGTs in GoM surface sediments. A) Ternary diagram showing the tetra-, penta-, and hexamethylated brGDGTs in GoM surface sediments plotted together with the global soil and peat dataset (Dearing Crampton-**
460 **Flood et al., 2019), b) the #rings$_{tetra}$, an indicator for marine produced brGDGTs and c) the Isomer Ratio (IR), indicative for fluvially produced brGDGTs.**

Despite the sedimentary in situ contribution to the total pool of brGDGTs in this zone, the total brGDGT concentration is substantially lower than the crenarchaeol concentration, which results in a low BIT index. Close to the MR, where BIT index values are >0.3, the #rings$_{tetra}$ ranges between 0.38-0.40. Although this is still slightly higher than the #rings$_{tetra}$ from soils in the catchment area of the MR, the #rings$_{tetra}$ close to the MR is substantially lower than elsewhere on the shelf, suggesting that the brGDGTs deposited close to the MR are predominantly soil-derived. Studies on the Yangtze, Tagus/Douro and Berau rivers have also reported high #rings$_{tetra}$ in shelf regions (Zhu et al., 2011; Zell et al., 2015; Sinninghe Damsté, 2016). In these studies, a high #rings$_{tetra}$ corresponded to rather low BIT index values, whereas in locations where the BIT index was high, a lower #rings$_{tetra}$ was observed, indicating that the contribution of in situ produced brGDGTs on the BIT index was trivial.

Fluvially produced brGDGTs can be identified by the relative abundance of 6-methyl brGDGTs, quantified with the IR. This is based on the finding that the IR can be higher for brGDGTs in river suspended matter and sediments than in those in the catchment soils. This is likely due to the pH of river water that is generally higher than that of soils, thus promoting the production of 6-methyl brGDGTs in river waters that will also transport soil-derived GDGTs (De Jonge et al., 2014a). For example, brGDGT-based pH estimates for suspended particulate matter in the Yenisei River were much higher than that of the surrounding soils, which indicates an in situ contribution of brGDGTs, and was reflected by high IR values of >0.7 (De Jonge et al., 2014a).

While the pH of MR water is 8.2 (data from USGS at St. Francisville, Louisiana https://www.epa.gov/waterdata/water-quality-data#legacy), pH values of soils in the river catchment range from 5-8 (Slessarev et al., 2016). With the generally higher pH in the river compared to soils, fluvially produced brGDGTs are thus expected to have a higher IR. However, soils in the Mississippi catchment have an average IR of 0.43 ± 0.15 SD, n=14 (soil pH = 6.20 ± 0.86 SD) (De Jonge et al., 2014a), which is similar to the range in the Mississippi transect (0.49-0.55, Fig. 10c). Therefore, a freshwater contribution to the brGDGT pool in these sediments seems insignificant. The high IR in deeper (>3000 m) waters is unlikely to be the result of river input but rather by an increased amount of brGDGT IIIa′ (Xu et al., 2020).

Close to the MR the brGDGT concentration and the BIT index are higher (>0.3) than in all other sediments. Since lacustrine in situ contribution is not evident, this indicates that the BIT index provides a realistic reflection of the fate of SMOM in the northern GoM. The stark decrease in the BIT index in the off-shore transects indicate that most of the soil-derived brGDGTs are deposited in the sediments close to the river mouth. If we can use brGDGTs as a tracer for the total pool of SMOM as a whole, this suggests that SMOM is not transported far after discharge. However, the BIT index is based on relative abundances of marine and terrestrial OM contributions, and a large contribution of marine OM may mask the terrestrial component. Normalizing the concentration of brGDGTs on TOC, however, reveals the same spatial pattern as that of the BIT index (Fig. 3), implying that soil-derived brGDGTs are probably not transported far into the marine realm, and are either directly transferred to the sediment or are rapidly lost after discharge into the GoM.

The BIT index marks a starker decrease of SMOM after discharge than the F$_{C32\,1,15}$ that reflects fluvial OM, although the spatial pattern of the F$_{C32\,1,15}$ resembles that of the BIT index (r = 0.71, p <0.005) and the brGDGT concentration (r = 0.89, p<0.005, Fig. 3 and 5). If we assume that soil-derived brGDGTs and river-produced diols are appropriate tracers for SMOM and fluvial

OM as a whole, this suggests that fluvial OM may be transported further onto the shelf than SMOM. SMOM is generally assumed to form associations with mineral surfaces, which would protect this OM from degradation during land-sea transport (Mayer, 1994a, b; Keil et al., 1997). However, several studies have shown that TerrOM appears to be progressively lost from mineral surfaces further offshore (Keil et al., 1997; Hou et al., 2020). During the transition from river to ocean, TerrOM might be replaced by marine OM due to the higher concentration of ions in seawater. Furthermore, TerrOM might be subjected to several cycles of deposition and erosion, causing TerrOM to transition between particulate and dissolved phases (Middelburg and Herman, 2007), which results in a net removal of TerrOM from mineral surfaces. At this freshwater/saltwater interface, flocculation of clay minerals may occur, which results in the sedimentation and thus loss of TerrOM out of the water column (Sholkovitz, 1976). However, the brGDGT concentrations decrease more rapidly than other TerrOM sources. Possibly, due to the hydrophobic nature of brGDGTs, they could potentially form colloids rather than fully moving to the dissolved phase (Kirkels et al., 2022). Studies on brGDGTs in rivers have shown that their composition is stable in suspended matter throughout the water column, i.e. there is no preferential loss or production of a certain brGDGT compound, whereas the grain size and mineral composition are clearly depth segregated (Feng et al., 2016; Kirkels et al., 2020). This suggests that either their bonds with mineral surfaces are continuously renewed during transport, or that they are present in the form of colloids. After discharge, these colloids might then stay in the seawater or degrade, which would explain their limited dispersal in the GoM sediments.

### 5.2.3 Higher plant-derived OM

We used long chain *n*-alkanes, pollen and spores in the surface sediments to trace higher plant OM input into the GoM. Here, we also discuss the potential use of stigmasterol, campesterol and β-sitosterol as indicators for higher plants, as these are also produced by higher plants (Volkman, 1986; Lütjohann, 2004). The plant markers are most abundant close to the MR mouth, where the highest concentrations of *n*-alkanes and pollen grains can be found, and on the shelf, where sterol concentrations peak (Fig 6, Fig. 7). The higher concentrations of higher plant markers on the shelf, matches that of the low $\delta^{13}C_{org}$ values (<-23‰) in this zone (Fig. 2c), suggesting that higher plant material may provide a substantial contribution to the TerrOM pool buried in the surface sediments. Further from the river mouth, plant markers decrease in abundance, consistent with the trends reported in earlier studies (Bianchi et al., 2002; Waterson and Canuel, 2008; Sampere et al., 2008). Despite their shared higher plant source, the different plant OM tracers show a distinct spatial pattern. For example, *n*-alkanes are present on the entire shelf transect (Fig. 7a), while pollen grains are most abundant close to the MR and on the westernmost shelf (Fig. 7b), where the proximity of extensive marshlands and river transport to Galveston Bay (Texas) might represent a secondary source for pollen. Although the exact contribution of marsh vegetation to the plant OM pool in the shelf sediments cannot be quantified, our pollen data indicate that marsh taxa occur in higher relative abundances close to shore (~20%), although their total concentration remains overall low. Notably, marsh plant pollen in the shelf sediments could also be introduced by the MR, which has been suggested as the primary source of pollen, and likely TerrOM, on the Louisiana shelf (Chmura et al., 1999). Furthermore, the ACL of *n*-alkanes does not show a spatial trend, suggesting that their source composition remained

unchanged. The presence of *n*-alkanes has also been observed in the central GoM, approximately 500 km south of the MR
mouth (Tipple and Pagani, 2010), indicating that these biomarkers can be transported for long distances. Pollen and *n*-alkanes
can be transported by both fluvial and aeolian transport pathways, but due to the large input of sediments and freshwater from
the MAR into the northern GoM, we assume that the MAR is likely the dominant mode of transport of *n*-alkanes and pollen
coming from the mainland US.

The relative abundance of stigmasterol, campesterol and β-sitosterol also varies spatially in the GoM; the fractional abundance
of stigmasterol and campesterol is highest in shallow sediments close to the MR mouth and on the shelf in the reach from the
river plume, whereas β-sitosterol is more dominant towards the open ocean (Fig. 6b,e,f). Since these sterols have additional
microalgal sources, a critical evaluation of their origin is needed. β-sitosterol is the most dominant sterol in higher plants
(Lütjohann, 2004), and in areas receiving large TerrOM inputs, it usually has a predominant higher plant origin (Yunker et al.,
1995). In the northern GoM, β-sitosterol is consistently the most abundant higher plant-associated sterol in the sediments.
Waterson and Canuel (2008) compared the concentrations of stigmasterol to brassicasterol in the northern GoM, to differentiate
between terrigenous and algal OM sources. They found that stigmasterol concentrations on the shelf correlated well with
stigmasterol concentrations in river and marsh samples, suggesting that this sterol was derived from vascular plants.
Nevertheless, the PCA performed on all biomarker, pollen, and dinoflagellate cyst concentrations from our samples shows that
all sterols have a low score on PC1 and plot in between the terrestrial and marine groups (Fig. 9). The similar response of all
sterols indicate that they have a common source, although the presumed higher plant-derived sterols plot on the 'terrestrial
side' of PC1, and the sterols derived from marine algae in the direction of other marine markers. Nevertheless, the different
PCA results for pollen, *n*-alkanes and these sterols imply that the here considered sterols include an substantial algal
contribution and are not solely derived from higher plants. Thus, pollen and *n*-alkanes may be more robust tracers of plant
material input in the northern GoM.

The spatial distribution of plant markers is also clearly different from those representing SMOM and freshwater OM, which
were constrained to the MR mouth and nearby shelf (Fig 3, 5). Possibly, the recalcitrant nature of plant OM might play a role,
as *n*-alkanes are likely less sensitive to microbial degradation than diols or brGDGTs. For example, a study on biomarkers in
turbidites found that n-alkanes were more resistant to oxic degradation than other biomarkers, including diols, sterols, fatty
acids and alcohols (Hoefs et al., 2002). Similarly, lignin phenols are also considered to be relatively resistant against
degradation (e.g., Hedges and Mann, 1979; Zonneveld et al., 2010) and show similar distribution patterns in the GoM as the
*n*-alkanes (Goñi et al., 1997; 1998; Bianchi et al., 2002; Sampere et al., 2008). This resistance may contribute to the observed
preferential transport of *n*-alkanes further into the coastal zone, compared to other TerrOM types. On the other hand, it has
also been shown that decay rates of lignin were similar to those of less recalcitrant organic compounds in the same soils and
based on $^{13}$C and $^{14}$C dating methods, this lignin also did not appear to be older than these compounds (Lützow et al., 2006;
Heim and Schmidt, 2007; Dynarski et al., 2020). In addition, earlier studies have suggested that the transport of plant markers
is sensitive to hydrodynamic processes in the coastal zone due to their association with mineral particles. Molecular plant
markers associated with clay particles can be transported further into the open ocean than plant remains, which are usually

associated with coarser grains (Goñi et al., 1997, 1998). The spatial distribution of plant-OM in the GoM shows that, in particular, *n*-alkanes could be preferentially associated with clay particles and thereby transported further off- and along shore, as these mineral associations not only facilitate their transport, but also provides protection from degradation (Mayer, 1994a, b; Keil et al., 1997; Lalonde et al., 2012). Possibly, the plant markers have a higher affinity to bind with mineral surfaces than $C_{32}$ 1,15-diols or brGDGTs, which likely do not form these associations (Kirkels et al., 2022). To support this, mineral associated organic matter in the soil continuum generally consist of more plant-derived OM than microbial-derived OM, although this depends on many factors including substrate quality, land use and climatic conditions (see Angst et al., 2021 for a review). Therefore, plant OM might be transported by sorption to mineral surfaces, explaining their more widespread occurrence and better preservation in the GoM.

## 5.3 Identification of specific niches for OM production

The previous sections illustrated that the composition of TerrOM varied substantially spatially. In this section, we investigate patterns of marine productivity and assess marine OM composition in surface sediments of the northern GoM and relate it to the different producers. Information on marine productivity can be useful in at least two ways. It helps evaluating the reliability of indices and proxies that are based on the relative contribution of the two types of OM, terrestrial and marine, such as the BIT and the C/N ratio. Additionally, the identification of the main producers of the marine OM at a certain location provides indication of the spatial distribution of those organisms in the modern system. Such information, applied to downcore sediments, allow to assess changes in the niches of the different marine producers through time. For this scope, we used various biomarkers and dinocysts. The marine markers (i.e., crenarchaeol, 1,14-diols, dinosterol) show the highest concentrations in the sediments of the shelf and the shallow part (<100 m water depth) of the Atchafalaya transect (Fig. 4). This pattern fits with the westward direction of the river plume over the shelf, confirming that marine production is triggered by the nutrients discharged from the MAR (e.g., Lohrenz et al., 1997). The high $\delta^{13}C_{org}$ and relatively low C/N ratio values on the shelf (Fig. 2) confirm substantial marine productivity and its contribution to sedimentary OM in this area. Marine production is relatively low around the mouth of the MR. This can be explained by the high sediment load of the MR that causes turbid waters, generating unfavourable conditions for marine primary producers as photosynthesis is limited by light penetration into the water column (Lohrenz et al., 1990; Wysocki et al., 2006).

The spatial distribution of the different marine productivity and OM markers in the sediments varies. For example, crenarchaeol produced by ammonia oxidizing Thaumarchaeota, is most abundant on the shelf (Fig. 4a), where nutrient concentrations are highest due to discharge by the MAR (Rabalais et al., 2002; Van Meter et al., 2018). This is also evident from the low C/N values (≤10) on the shelf that indicate a nitrogen surplus (Fig. 2c). The long-chain 1,14-diols also mainly occur in the shelf sediments under the influence of the river plume (Fig. 4b). These compounds are produced by *Proboscia* diatoms, which thrive in high-nutrient areas (Sinninghe Damsté et al., 2003), where silica brought by the river is abundant. Accordingly, previous studies show that diatoms are the dominant phytoplankton group on the shelf (Wysocki et al., 2006; Liu and Xue, 2020). In contrast to crenarchaeol and diols, alkenones are most abundant in sediments at a water depth of 30-300 m, further offshore

(Fig. 4c). This agrees with the niche for alkenone-producing haptophytes that are mostly observed in more oligotrophic, less turbid waters (e.g., Menschel et al., 2016) and hence occur further offshore than the diatoms. In the southwestern GoM, haptophytes were most abundant in the deep photic zone (50-100 m water depth, Baumann and Boeckel, 2013).

The marine sterols brassicasterol and dinosterol are again most abundant on the shelf, although their relative abundances show different patterns, where dinosterol is relatively more abundant in deeper waters compared to brassicasterol (Fig. 6c-d). Brassicasterol is often a predominant sterol of diatoms (Rampen et al., 2010), although it has also been attributed to many groups of marine phytoplankton (e.g., Volkman, 2003). Dinosterol is primarily produced by dinoflagellates, both the autotrophic and the heterotrophic ones, but its production can vary remarkably depending on the species (Volkman et al., 1993; Amo et al., 2010). The relationship between dinosterol abundance and dinocyst concentrations is complex (e.g., Leblond and Chapman, 2002; Mouradian et al., 2007), since not all dinoflagellates make cysts and dinosterol is not produced by all dinoflagellates. This is illustrated here by the total dinocyst concentrations, which are, in contrast with the distribution of dinosterol, higher on the shelf and decrease towards the open ocean.

Dinocysts also allow us to zoom in on the food chain, as autotrophic and heterotrophic dinocysts represent primary and secondary producers, respectively. The autotrophic species are clearly dominant in sediments of the GoM outside the river plume, i.e., towards the open ocean, and sediments along the Atchafalaya transect, where the water depth is >80 m (Fig. 8). Here, waters are less turbid, and light can penetrate deeper. The heterotrophic taxa are not dependent on light availability but exclusively on OM availability for food (e.g., diatoms and other organic debris) and dominate on the shelf and the shallow parts of the Atchafalaya and Mississippi transect. Other studies on dinocyst assemblages in surface sediments in the GoM also find high dinocyst concentrations and a strong dominance of heterotrophic species on the Louisiana shelf (Price et al., 2018), while dinocyst further offshore are less abundant and the assemblages are generally dominated by autotrophic species (Limoges et al., 2013).

As stated above, the amount of oxygen is bottom waters and sediments plays a role in dinocyst preservation. Previous studies have shown that in well oxygenated waters, heterotrophic cysts are more prone to degradation than autotrophic ones (Zonneveld et al., 2010). Since our surface sediment dataset is partly located in a seasonally hypoxic zone, but also includes surface sediments from water that are well oxygenated year-round, the high percentages of heterotrophic species on the shelf could be biased by the seasonally lower oxygen concentrations. However, several studies have demonstrated that the concentrations of heterotrophic cysts do correspond with export productivity (Reichart and Brinkhuis, 2003; Zonneveld et al., 2009). Dinocyst (and heterotrophic dinocyst) concentrations in the GoM are highest close to shore, indicating that the concentration signal is not biased by differential preservation and still shows heterotrophic dinocyst dominance there.

The dominance of heterotrophs close to the MR in combination with the limited transfer of SMOM and river-derived OM after discharge suggests the occurrence of priming. The priming effect is based on the increased turnover of recalcitrant OM upon addition of a more labile OM-source (Löhnis, 1926; Kuzyakov et al., 2000). In the coastal marine environment, the addition of freshly produced algal OM to supposedly more refractory TerrOM may trigger an increase in the decomposition rate, whereby also the more recalcitrant fraction is remineralized (Guenet et al., 2010; Bianchi, 2011). Priming has not been widely

studied in marine systems, but several studies have shown increased TerrOM mineralization after the addition of primary produced OM (e.g., van Nugteren et al., 2009; Gontikaki et al., 2015), although results on the magnitude of the priming effect in aquatic systems are inconsistent (Bengtsson et al., 2018; Gontikaki and Witte, 2019). In the GoM, previous studies have shown correlations between the concentrations of phytoplankton populations (mainly diatoms) and POC concentrations in the plume direction of the MR, suggesting that the MR discharge had a strong influence on the phytoplankton concentrations

(Wysocki et al., 2006). Furthermore, degradation products of chlorophyll-*a* indicative for zooplankton grazing were present on the Louisiana shelf (Chen et al., 2003), suggesting that the Louisiana shelf is a hotspot for both marine and terrestrial OM degradation. Chin-Leo and Benner (1992) estimated that in the MR plume, heterotrophic bacteria, often considered the major consumers of dissolved OM in aquatic systems, were unable to sustain their respiration by consuming phytoplankton biomass alone, and that river-discharged TerrOM sources must have been also utilized. On the other hand, degraded TerrOM was

mostly found in bottom waters of the northern GoM rather than in surface waters, suggesting that in surface waters, labile, marine-sourced OM is likely preferably degraded instead of more the resistant TerrOM (Liu and Xue, 2020).

In our dataset, the occurrence of priming would imply that especially SMOM and freshwater-derived OM is remineralized in the presence of fresh marine OM, which could explain the limited dispersal of SMOM and fluvial-derived OM after discharge. However, since we did not perform isotope labelling experiments to detect priming, we cannot exclude that differences in the

mineral protection or lability of SMOM and fluvial-derived OM as opposed to plant-derived OM discussed earlier may be the main factors that explain the spatial distribution patterns observed here.

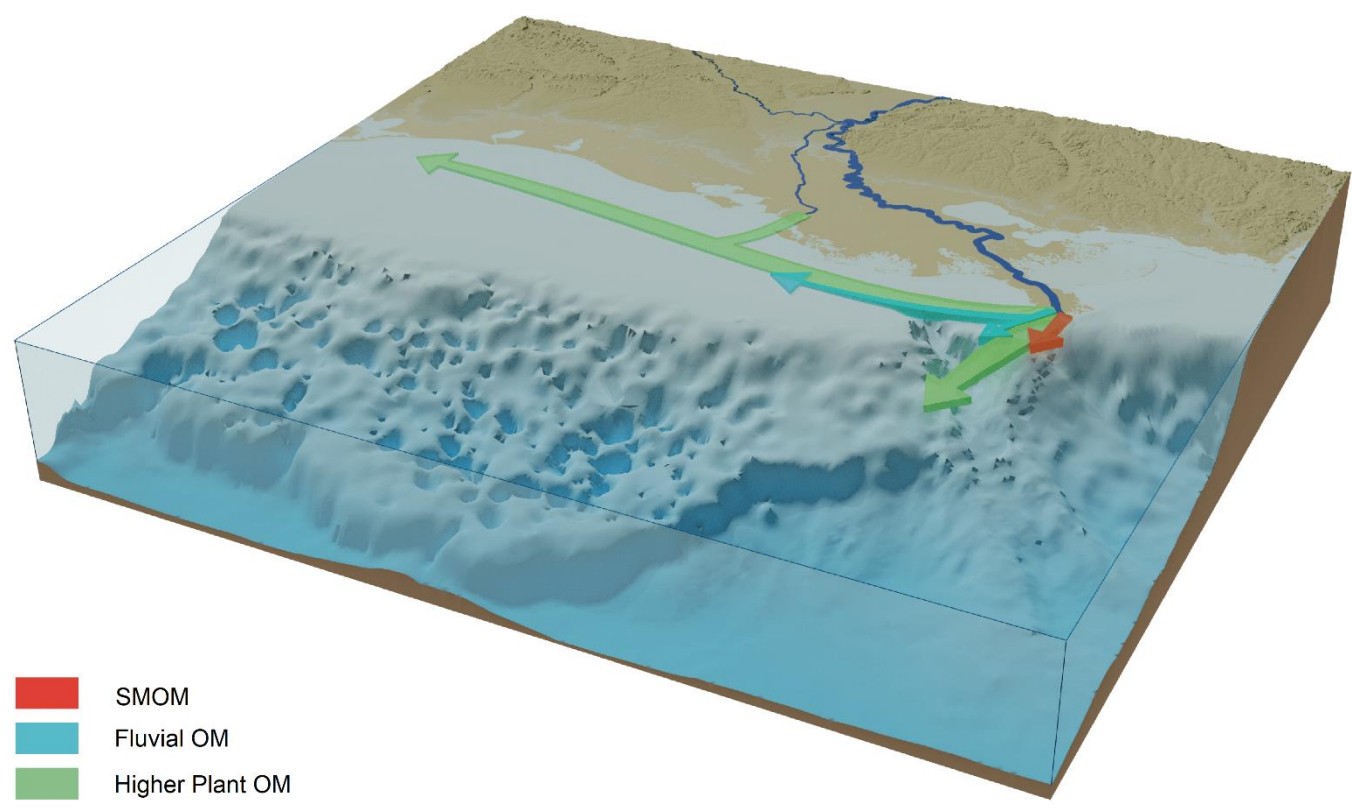

| | SMOM |
| | Fluvial OM |
| | Higher Plant OM |

**Fig 10. Schematic overview of the dispersal of SMOM, fluvial- and higher plant-derived OM in the northern GoM based on the lipid biomarker, pollen, and dinocyst distributions from this study.**

## 6 Conclusion

The biomarker and pollen distributions reveal that different types of TerrOM have distinct dispersal patterns upon discharge in the coastal zone (Fig. 11), where plant material reaches the deepest waters, while fluvial- and especially SMOM are more confined to the proximity of the MR mouth. The distinct dispersal patterns of these different types of TerrOM can be linked to specific transport mechanisms and degradation processes for these TerrOM pools. SMOM seems to be rapidly lost after

655 discharge, either by forming colloids and subsequent dispersion in the water column or degraded due to priming, whereas plant markers are found further offshore likely associated with mineral surfaces, which protects them from degradation and facilitate their transport. The spatial variation in composition of the TerrOM in the GoM sediments implies that also the efficiency of coastal marine sediments as a long-term carbon sink is spatially variable. This has implications on the role of coastal zones as a hotspot for OM-burial, where mostly plant material has a higher burial efficiency on continental shelves.

Furthermore, the concentration of marine biomarkers and dinocysts in the surface sediments indicate that marine OM production is highest on the shelf. The spatial distribution of the biomarkers in the surface sediments reveals the preferred niches of the different marine producers: dinoflagellates, diatoms, and Thaumarchaeota are found close to the coast, while

coccolithophores are more present further offshore. In general, the shelf community is dominated by heterotrophic dinoflagellate species, whereas autotrophic species occur in more oligotrophic and/or less turbid waters further offshore. The dominance of heterotrophic dinocysts close to the MR mouth suggest that priming may have led to increased degradation of recalcitrant, mostly soil-microbial TerrOM, limiting its dispersal further offshore. Our study thus shows that a detailed investigation on the composition of OM in marine continental margin sediments can reveal the processes that drive TerrOM distribution and burial in the marine environment.

**Data statement**

The dataset from this study is accessible through Pangaea (https://doi.pangaea.de/10.1594/PANGAEA.944838 (Yedema et al., 2022)). The global soil and peat dataset is available at https://doi.org/10.1594/PANGAEA.907818 (Dearing Crampton-Flood et al., 2019).

**Author contributions**

FP and FS designed this study. FP and FS led the sample collection and supervised YWY, who generated the data, performed the data analyses and wrote the manuscript together with all co-authors. All co-authors contributed to the data interpretation and provided feedback on the manuscript.

**Competing interests**

The authors declare that they have no conflict of interest.

**Acknowledgements**

We thank two anonymous reviewers and the editor for their comments and suggestions to improve this manuscript. This work was carried out under the program of the Netherlands Earth System Science Centre (NESSC), financially supported by the Dutch Ministry of Education, Culture and Science (OCW) (NESSC Gravitation Grant; grant no. 024.002.001). We thank Klaas Nierop, Desmond Eefting, Arnold van Dijk, Natasja Welters and Giovanni Dammers for their technical support and Timme Donders for help with the identification of pollen (all Utrecht University). We also thank the scientists and crew aboard of Pelagia cruise 64PE467.

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
