# Peer review of "The dispersal of fluvially discharged and marine, shelf-produced particulate organic matter in the northern Gulf of Mexico"

_Biogeosciences, 2022_

## Author Comment (AC2)

**Reply to reviewer #2:**

The manuscript entitled "The dispersal of fluvially discharged and marine, shelf-produced particulate organic matter in the northern Gulf of Mexico" by Yedema and coauthors takes advantage of the combination of organic geochemistry and palynology to determine the origine of sedimentary organic matter and ecological niches in the northern Gulf of Mexico.

The description of geochemical data is complete and discuss the relative abundance of GDGTs, long chain diols and, more classical for this type of study, n-alkanes, alkenones and sterols, leading to assumption on the fate of soil derived, fluvialy derived and marine OM.

This is a nice descriptive paper, well-written (even if, as a non-native english speaker, I can not evaluate the use of English language). My only concern is about the apparent lack of a scientific question. The issue of the fate of terrestrial OM at the land-sea interface is, of course, important for C cycling, but the introduction could indicate why and how the present results will contribute to create new knowledges. Assumptions are formulated to discuss the relative preservation of the different sources of terrestrial OM, then there is just to make this point explicit.

Finally, in order to be more efficient on the use of money and energy, I suggest for further research that, at this stage of the knowledges on the northern Gulf of Mexico system, it could be interesting to test hypotheses on the fate of OM with a specific sampling strategy or using experimentations instead of formulating assumptions after the sampling.

This article is close to being published after minor revisions.

*Reply: We thank this reviewer for the positive assessment of the manuscript and praise on its completeness regarding the geochemical data.*
*Regarding the specific comments:*
*-Research question: In this study, we aim to identify the TerrOM composition stored in the northern GoM to assess if specific OM pools are preferentially buried in the sediments. To further motivate our aim, we will add more information on previous biomarker work in the GoM to the introduction and specify the knowledge gaps that remain. We will also further clarify how our dataset will contribute to closing this knowledge gap.*
*- Study design: Studying the dispersal of TerrOM in the northern GoM was the main objective when planning the research expedition. We have deliberately collected sediments along two land-sea transect starting at the river mouths of the AR and the MR, as well as one shelf transect that follows the MAR river plume to trace TerrOM. We will clarify this further in the revised manuscript.*

---

## Author Response (AR1)

Dear editor, dear Ji-Hyung Park,

Thank you for your positive evaluation of our manuscript and our replies to the referees' comments.

We have revised the manuscript based on your and the reviewers' comments. The main changes to our manuscript are:
- an extension of the introduction. We now include more of the previous biomarker work on the Louisiana shelf as suggested by reviewer #1. We also clarified the novelty of our work considering the findings of these earlier studies, and added references to these studies where relevant in the manuscript. We have also rephrased our research aim and scientific question. (Lines 85-95).

- an extended explanation of the role of hypoxia on the production and fate of OM in shelf sediments in section 3.1, as requested by reviewer #1 and yourself. In short, we give an overview of the seasonal variability of hypoxia in the GoM and how it possibly influences the fate of OM. This section now also addresses the differences in bathymetry between the MR and AR shelf areas, as well as the influence of the Loop Current on the Louisiana shelf.

- An explanation of the influence of (seasonally) changing oxygen conditions on our surface sediments in section 3.2 to address comments of reviewer 1. (Lines 226-240).

- an addition of the discussion on the variation of transport of different plant biomarkers on request of reviewer#1. We have added the suggestion that the more resistant nature of *n*-alkanes compared to diols and brGDGTs could contribute to the distinct dispersal patterns of the different TerrOM pools (Lines 563-568). Furthermore, we briefly discuss the possible contributions of OM from marshes, for which we do not find evidence in our samples and thus consider insignificant compared to the input via the MAR (Lines 529-532).

- Regarding the editor's comment about the possible confusion with differentiating between 'soil' OM and 'plant' OM, we have now specified brGDGTs as soil microbial OM (SMOM) instead of soil-derived OM throughout the manuscript. All figures and figure captions have been updated accordingly.

- A more elaborate discussion on the possible occurrence of priming in the northern GoM, supported with findings from the literature. Furthermore, we discuss the differences of SMOM, plant OM, and fluvial OM in relation to priming and the possible influence of their sensitivity to degradation due to their molecular structure or protection by mineral associations (Lines 625-646).

A point-to-point list of detailed changes in response to comments of the referees and the editor can be found below. The changes in the revised manuscript are made with track changes on.

We hope that you find this revised version suitable for publication in Biogeosciences.

On behalf of all co-authors,
Yord Yedema

**Here below we added:**
1) **A detailed reply of the comments raised by the editor,**
2) **The comments of reviewer #1 and #2 with our reply (in blue) and the changes made**

**Editor comments**

Above all, you need to clearly state your objectives and hypotheses based on an overview of prior OM and biomarker studies on the GoM.

*Changes made: We extended the overview of prior biomarker studies on the GoM in the introduction and have rephrased our objectives to clarify that our study will investigate the dispersal of multiple terrestrial OM sources (i.e., plant, soil microbial, and fresh water) in the northern Gulf of Mexico after discharge.*

Regarding your response to the first reviewer's comment on the significance of hypoxia, please go beyond indicating the observed oxic conditions during your one-time sampling to provide a more in-depth discussion of seasonal variability in oxygen availability and its impacts on the different fates of terrestrial vs. marine OM.

*Changes made: As mentioned in the general list of changes, we have extended the text on the role of hypoxia on the fate of terrestrial vs marine OM in section 3.1 and added a motivated assumption on its influence on the OM in the surface sediments studied here.*

Given the importance of accurate organic analyses in your study, I would like to ask you to provide more details on QC results. For instance, what lab standards you used for both elemental and isotope analyses, how you determined the analytical errors, etc.

*Changes made: We added the names of the standards that were used for bulk sediment elemental and isotope analysis, as well as what the analytical errors are based on to the methods.*

As a 'terrestrial' biogeochemist, I had difficulty differentiating between the 'soil-derived' OM and the 'plant-derived' OM. As you know, the bulk of SOM is actually derived from plants. From your response to a reviewer comment, it appears that you focus on the OM of soil microbial origin (involved in decomposition and humification) for your soil markers. It will be helpful to define the term in a proper place.

*Changes made: We agree with this comment, and now refer to soil microbial OM (SMOM) instead of soil-derived OM when interpreting brGDGT data.*

Lines 511-513 (plant markers have a higher affinity to bind with mineral surfaces than brGDGTs or C32 1,15-diols,,,): Is this simply related to the size or any chemical characteristics such as the difference in surface charges?

*Changes made: The affinity to bind with mineral surfaces may indeed be related to the chemical properties of the molecule. We have extended the discussion on this topic, and now present the different mechanisms that may explain the observed dispersal patterns in a more speculative way. See lines 553-568.*

This may be a side issue, but I wondered whether the very general term 'multi-disciplinary' approach would best describe your approach employing multiple biogeochemical indices.

*Changes made: Since palynology and organic geochemistry are generally separated fields, we thought that the term multi-disciplinary was appropriate here. Nevertheless, we changed 'multi-disciplinary' to 'combined lipid biomarker and palynology'.*

**Reviewer #1**

1) Since surface sediment composition could vary through seasons, timing of field sampling and oxygen availability are critical for interpreting OC-sources and preservation. However, this manuscript neglected the significant impact of hypoxia by providing the reason that water may become reoxygenated in the next season. In fact, the systems may not be completely reset, and this seasonal redox oscillation itself could also enhance or retard OC degradation by various mechanisms.

*Reply: Surface sediments are here represented by the upper 2 cm of multicore sediments. We do have $^{210}$Pb dates for several of these samples. The highest sedimentation rates found in these 2 cm is ~1cm/year in the proximity of the Mississippi river. Consequently, the surface sediments presented in this manuscript represent at least one year of organic matter deposition, therefore integrating seasonally varying oxygen conditions. Moreover, sampling was carried out in February 2020, and all sampled locations had oxic water conditions at that time. We will include this information in the revised version of our manuscript.*

*Changes made: We have done this, and also extended the text on the general influence of hypoxia on OM preservation in the northern GoM as also suggested by the editor (see section 3.1).*

2) The discussion of priming mechanism is still obscure. There was no explicit evidence to suggest that decomposition of soil-derived OC was boosted by addition of labile materials. Moreover, the authors should provide more evidences to support why priming of OC decomposition selectively affected soil-derived OC, but not influenced plant-derived OM.

*Reply: We agree with the reviewer that we have no direct evidence of priming in the coastal zone. Priming is hard to directly measure in natural systems, especially without using isotope labeling. Hence, we here infer priming from our observation that offshore TerrOM transport is limited and coincides with a high amount of heterotrophic dinocysts close to shore, suggesting an enhanced activity of secondary producers, as described in the manuscript (lines 549-566).*

*We also do not have direct evidence that soil-OM is preferentially targeted by priming per se. However, based on the distinct dispersal patterns of the TerrOM types, we hypothesize that certain pools of OM (such as plant-derived OM) may be protected by mineral associations (see Lines 511-513), consistent with previous observations (e.g. Repasch et al., 2022, Geophys. Res. Lett; Kirkels et al., 2022 Biogeosciences). We will extent the discussion of the priming process in the revised manuscript with an emphasize on the difference between soil and plant TerrOM.*

**Changes made: We have done this, also following the request of the editor. See extended discussion in section 5.3.**

**Line 44-46:** "initial composition of this particulate OM influences the burial efficiency of TerrOM" The discussion on "burial efficiency" requires incorporation of other data that used cores and biomarkers in this region and the issues of hypoxia. Please look at the following papers, and references therein, that I think should prove useful: Bianchi et al., 2002 Mar. Chem. 77: 211-223; Chen et al. 2003 GCA.: 67: 2027-2042; Chen et al. 2003 Mar. Chem.,: 81: 37-55; Bianchi et al., 2006 Eos: 87 (50): 565, 572-573; Bianchi et al., 2007 Estuar. Cast Shelf Sci., 73: 211-222; Bianchi et al., 2007 GCA: 71: 4425-4437; Sampere et al., 2008 Cont Shelf Res. 28: 2472-2487; Bianchi et al., 2010: Sci. Total Env. 408: 1471-1484; Sampere et al 2011 Estuar. Coastal Shelf Sci. 95: 232-244.

*Reply: We thank the reviewer for these suggestions. We will take a careful look at the above-mentioned papers and include references in the revised manuscript where relevant.*

**Changes made: We reviewed the abovementioned literature and included relevant papers to our revised manuscript. Specifically, the suggested literature was used to extend the introduction of previous biomarker work in the northern GoM (lines 71-88), as well as study site description and the discussion on plant material and priming (section 5.2.3 and 5.3).**

**Line 115-117:**

1) Is "ammonium oxidizer" more commonly used than "ammonia oxidizer"?

*Reply: Ammonia oxidizer is indeed the more commonly used term. We will change this in the revised version of the manuscript.*

**Changes made: We now consistently use ammonia in our manuscript.**

2) Since Thaemarchaeota is an ammonia oxidizer, are their other papers form this region on their abundance as related to oxygen availability, not just ammonia concentration?

*Reply: Previous studies that monitored Thaumarchaeota populations in the northern Gulf of Mexico (e.g. Tolar et al., 2013; Front. Microbiol.) show that while Thaumarchaeota are present across the northern GoM, there is no clear relation with oxygen availability. Since the northern GoM is characterized by high primary productivity and subsequent high degradation of OC releasing ammonia, we believe that ammonia availability is a more important factor that influences the abundance of Thaumarchaeota than oxygen concentrations.*

**Changes made: None.**

**Line 133-134:** Seems like the n-alkanes data set would be more comprehensive if the authors added short-chained (C17-C19, C21) and mid-chained (C23, C25) n-alkanes as proxies of marine algae and aquatic macrophytes, respectively. This could be linked to some of the papers cited above that use algal biomarkers in this region.

*Reply: We chose to focus on the concentrations of n-C29-C35 to specifically target the terrestrial plant material as part of the TerrOM that is discharged by the Mississippi River. The short- and mid-chain n-alkanes occur in substantially lower concentrations than the long-chain n-alkanes, which is why we decided not to include the data in the manuscript. Regardless, Paq (Ficken et al., 2000) values are ≤0.3 for almost all samples, reflecting the dominance of long-chain n-alkanes and suggesting a predominant higher plant source over that of submerged/floating aquatic plants. Moreover, the short-chain alkanes represent only a very minor (<3%) portion of the total n-alkane pool at most sample sites but reach highest relative abundances close to the Atchafalaya River mouth (9-11% at site A15-A50), consistent with the high concentrations of the marine biomarkers. This shows that these short-chain n-alkanes indeed correlate with the marine algae here. As indicated above, we will take a careful look at the above-mentioned papers and include references in the revised manuscript where appropriate.*

**Changes made: We have added references to some of the above-mentioned paper in sections 5.2.3, 5.3, as suggested by the reviewer and explained above in the reply.**

**Line 180-182**: How might loop current seasonal variation matter? Check papers by Doug Biggs…

*Reply: In general, the Loop Current extents further north in the GoM during spring-summer months. However, the Loop Current is usually affecting water properties to the east of the Mississippi river, and rarely reaches as far north as our study sites (i.e., the Louisiana shelf), except in unusual conditions via detached warm water eddies. We can thus assume that the Mississippi-Atchafalaya River is by far the dominant factor at our site, but before resubmission we will check recent observations of the Loop Current to assess its potential influence on our sedimentary components. Earlier palynological studies have also indicated little influence of the Loop Current in our region (Limoges et al., 2013 Mar. Micropaleontol.; 2014 Palaeogeogr. Palaeoclimatol. Palaeoecol.).*

**Changes made: We have described the (limited) influence of the Loop Current on our study site to the study site description (section 3.1) also in view of findings of the papers Biggs et**

*al., 1996 and Hamilton et al., (1999). Smitz Jr. et al., (2005), Schiller et al., (2011), Schiller and Kourafalou et al., (2014), on the influence of the eddies. "There were no apparent preferred paths either in the main basin or near the western slope where eddies were equally likely to move northward or southward along the boundary. Eddy paths also showed frequent occurrences of 20- to 30-day anticyclonic perturbations similar to that found from the individual analyses". Moreover, these papers mainly refer to the high variability of the interaction between the Loop Current and the plume of the Mississippi River, by also considering the importance of the freshwater discharge taken up by the LC and further transported. The shelf area is relatively and mostly episodically untouched by the main LC.*

**Line 185-188:** The discussion could use more perspective on the differences in slope and particle export rates between MR and AR.  See McKee et al. 2004 Cont. Shelf Res. 24: 899-926.

*Reply: We believe that this section already touches upon the difference in slopes between the Mississippi River and Atchafalaya River, as we already explain that the particle export rate of the Mississippi River is higher compared to that of the Atchafalaya River (lines 184-189). However, we will add further details on the export towards the Mississippi canyon and the impact of hypoxia on our study sites based on the suggested literature.*

**Changes made: We added more detailed information on the difference in slope and particle export rates to section 3.1.**

**Line 199**: Surface sediments (0-2 cm) should be discussed in the context of known sedimentation and burial rates and periods of export (see citations above).

*Reply: We will do this. See our reply on comment 1.*

**Changes made: We have done this and added a reference to Lenstra et al., 2022, who determined sedimentation rates of ~1.5 cm/y for the shallowest shelf sediments, decreasing to 0.2-0.3 cm/y on the slope.**

**Line 207-208:** Please look at Bianchi et al 2010 paper on hypoxia that cites relevant physical mixing and hypoxia seasonality papers to better interpret the context of these biomarkers. For example, if sediment discharge and OC input is extremely high during summer hypoxia, rapid burial rate may push fresh OC deep down into sediments.

*Reply: We agree with this comment. Indeed, if discharge is high, hypoxia will facilitate the rapid burial of fresh OM. However, as we stated above, the $^{210}$Pb dating of our sediment cores indicates that at the time of sampling, the surface sediments are younger than the underlying sediment at our sampling locations. The upper 2 cm we selected contain at the least the last 2 years.*
*Regardless, we will add relevant references, including Bianchi et al., 2010 and Hetland and DiMarco (2008; J Mar Syst.) to our text to better outline the potential effects of hypoxia on OM burial rates close to both river mouths in the revised manuscript.*

***Changes made: We have extended the section on the development of hypoxia and the factors that influence this (section 3.1). Specifically, we have included sedimentary organic matter respiration on the shelf close to the Atchafalaya River as a contributing factor to the development of hypoxia, including relevant references.***

**Line 266**: for the whole palynological processing paragraph: From figure 7, the authors state that dinocyst counts were normalized to TOC., are pollen counts also normalized to TOC as well? For comparisons, it might be interesting to normalize pollen and dinocyst count by weight (or volume) of sediments, since they are part of the less reactive sedimentary OC pool, similar to what is done with sigma lignin.

*Reply: Generally pollen grains and dinocysts concentrations are indeed calculated and reported per gram sediment (weight). Nevertheless, we here chose to report the concentrations per gram TOC to enable a more direct comparison with biomarker data, as we explained in lines 161-169 of the original manuscript. The pollen/dinocyst concentrations per gram sediment and per gram TOC are both part of the datafile we submitted to the PANGAEA open-access database. Regardless of the normalization considered (per gram sediment or per gram TOC), the spatial trends are comparable. Only samples located on the western shelf (especially site 20f), deviate from the trend when using the two normalizations. As these sites are characterized by very low TOC values (0.05-0.5 wt.%) and larger grain size (sand) compared to the others, we think that normalization to TOC may even be a better way to present palynological data. In the following figure (not in the manuscript) we compare the dinocyst concentrations per gram sediment and per gram TOC, which indicates that the difference is solely caused by the low TOC values of the western shelf. This has not changed our interpretation of the dinocyst trends. As mentioned above, we use the concentrations per gram TOC here to enable comparison with our biomarkers.*

[Figure]

***Changes made: None. The figure clearly shows that the dinocysts are more abundant on the shelf and in the shallowest areas were enough nutrients are available and turbidity (which limits photosynthesis) is reduced.***

**Line 305:** Figure 4: I personally agree with the ideas that the authors classified proxies into 4 figures including soil-derived, fluvial-derived, marine-derived OC, and plant-derived OC. However, according to Line 137-144, the authors mentioned that most of these sterols (especially, β-sitosterol, stigmasterol, and sitosterol) can be derived from terrestrial sources as well. Moreover, "total sterols" do not really reflect specific terrestrial and/or marine sources, since it commonly includes a mixture of both. In order to avoid misconception, the authors could remove these "sterol proxies" from figure 4, and be added to another figure specifically

for sterol proxies. Once again, using 2 and 3 end-member stable isotopic mixing models previous published for this region should help ground the interpretations here.

*Reply: We thank the reviewer for their advice on the placement of the sterol proxies. We will make a separate figure with the sterol data for the revised manuscript and add clarifications to the text where appropriate.*

**Changes made: We have made a separate figure for the sterols and changed all figure numbers accordingly. We have also elaborated on the presumably mixed sources of the sterols in the discussion.**

**Line 310:** Figure 5: It appears that the C32, 1-15 diol was transported west in the Louisiana Current with very little export off shore, look at physical oceanography paper in this region by Steve DiMarco, Ron Hetland etc. This is different from the other biomarkers that show strong export trend both along shelf and across shelf (e.g., n-alkanes). Is there any difference in hydrodynamics between these biomarkers?

*Reply: This is indeed curious. We surmise that differences in transport pathways between biomarkers resulting from e.g., mineral sorption, plays a role here. Alternatively, the C32 1,15 diols (together with brGDGTs) are less resistant to oxidation compared to n-alkanes (see Hoefs et al., 200 GCA), which can limit their transport further offshore. In addition, the C32 1,15-diols plot in between the terrestrial and marine proxies in our PCA results, implying that it might be possible that in situ production of C32 1,15 diols takes place on the shelf, near sites A15 and 20b. On the other hand, the FC32 clearly shows high relative abundances of the C32 1,15-diol near the Mississippi river, indicating a mainly riverine source of these diols. We will include this discussion in the revised manuscript.*

**Changes made: This discussion is incorporated in the revised manuscript. See lines 563-568.**

**Line 315:** Add "total" to The highest "total" sterol concentrations… (The phrase "The highest sterol concentrations" alone may be misinterpreted that the concentrations of each individual sterol are all highest between MR and AR).

*Reply: Thanks for spotting this, we will add the word 'total' to this sentence for clarification.*

**Changes made: We have done this.**

**Line 347-350:**

"Almost all variables plot positively on PC1, together with shallow shelf (<20 m water depth) sediments. The only exception is the concentration of alkenones, which plot negatively on PC1, with sediments at intermediate water depth (<80 m) on the Atchafalaya transect. Sediments from the deeper parts of the Mississippi (>50 m) and Atchafalaya (>200 m) transects also plot negative on PC1."

Sediments were separated… "sediments at intermediate water depth (<80 m) on the Atchafalaya transect" from "sediments from deeper Atchafalaya transect", why? Were they both plotted negatively on PC1?

*Reply: In hindsight, this part of the text was rather confusing and we will clarify this in a revised version. But indeed, sediments from both parts of the Atchafalaya transect plotted negatively on PC1. They were first separated since the alkenones plotted close to site 80b, which we called intermediate depth here, while the remainder of the transect also plotted negatively, but not close to the alkenones. In the revised version we will abstain from using 'intermediate' and 'deeper part as a way of describing the transect in this section.*

**Changes made: We clarified the description of our PCA results. See lines 390-393.**

**Line 369**: Since δ13C are all in negative range, the authors may want to use the term "less negative" or "more enriched" rather than "more positive"

*Reply: We will revert to the consequent use of the conventional indications: (relatively) $^{13}$C-depleted vs $^{13}$C-enriched or higher vs lower $\delta^{13}$C.*

**Changes made: We have done this.**

**Line 394-395:** The plume of high concentration of C32 1, 15 diol is correlated with zone of δ13Corg enrichment. Is this evidence for enhanced marine productivity via fluvial export?

*Reply: We think that the less negative δ13Corg in this area is mostly caused by the high marine productivity at this site as revealed by high concentrations of marine markers (crenarchaeol, alkenones, 1,14 diols) and dinoflagellates. The increased marine productivity is probably triggered by nutrients supplied by the Mississippi. We interpret the presence of TerrOM at the same site as an indicator that this OM may contribute to marine productivity (see lines 525 - 529), as we later describe in the priming section.*

**Changes made: none**

**Line 459-463**: What's about fluvial OM, can sorption on mineral surface be important?

*Reply: This is a very interesting point. This dataset indeed raises questions on mineral protection but the present data does not allow for a detailed analysis on this aspect. We are currently investigating the sorption to minerals of soil-, fluvial- and plant derived OM on a land-sea transect in the GoM to follow up on such observations.*

**Changes made: None.**

**Line 468:** Can we use brGDGTs as a representative of soil-derived OM in term of sorption mechanism? brGDGTs may represent a small fraction of total soil-derived OM. Does the rest of soil-derived OM (e.g., humic substances which enriched in polar functional groups) share the same sedimentation pattern with brGDGTs?

*Reply: In this paper, we present brGDGTs as representation of soil-derived OM, but the long-chain n-alkanes, while derived from higher plant, can of course also be stored in soils prior to mobilization and be transported to the coastal zone. As brGDGTs and n-alkanes show different dispersal patterns, this suggests that the OM source (in this case soil microbial vs higher plant) is more important than the specific compartment (e.g. soil, vegetation, aquatic) of the river system that the OM is derived from.*

**Changes made: As also suggested by the editor, we now refer to soil microbial OM rather than soil-derived OM when interpreting and discussing our brGDGT data.**

**Line 500-502:** Alternatively, is it possible that the distribution of n-alkanes and pollen greatly represented terrestrial input because they were more resistant toward degradation. However, sterols are more enriched in reactive functional groups; thus, their spatial patterns were more irregular due to heterogenous conditions for degradation (e.g., oxygen availability, the presence of microbes etc.). As discussed in previous comment (Line 468), n-alkanes represent only one fraction of total plant-derived OM. Can we assume that the rest of plant-derived OM share the same behavior with n-alkanes?

*Reply: Other plant material might behave differently compared to n-alkanes. However, previous studies from the Gulf of Mexico show that lignin concentrations decrease further offshore (Bianchi et al., 2002 Mar. Chem; Sampere et al., 2008 Cont. Shelf Res; Sampere et al., 2011) or remain constant in waters >100 m deep (Goñi et al., 1998 GCA). Another study that compared offshore trends of n-alkanes and fatty acids found similarly decreasing trends of both biomarkers (Hou et al., 2020 J. Geophys. Res. Biogeosci.). Nevertheless, lignin and n-alkanes both represent a resistant part of the plant OM-pool and might therefore be transported further than less resistant plant material (Hoefs et al., 2002 GCA). Therefore, it is possible that different types of plant-derived OM have different dispersal patterns.*
*Comparison of trends in n-alkane concentrations with those of sterols and pollen might not be totally fair due to the mixed sources of sterols in the GoM and the likely different transport mechanism of pollen, respectively. Regardless, we will add this discussion to section 5.2.3 of our manuscript.*

**Changes made: We included a discussion on the differences in transport between n-alkanes and other plant proxies in section 5.2.3. Specifically, n-alkanes and lignin represent a relatively resistant pool of OM and might therefore be transported further into the ocean. However, other studies on the GoM also that lignin phenols and fatty acids decreased offshore in a similar fashion (Sampere et al., 2008; Waterson and Canuel, 2008), suggesting that although differences in transport between plant proxies may exist, they seem to be transported further offshore than other TerrOM markers.**

**Line 505-507:** Is there any difference in sorption mechanism of soil-derived, fluvial-derived, and plant-derived OM on mineral surface? (For example, type of minerals, particle size, and etc.), see paper by Mayer et al., 2009 Mar Chem.

*Reply: We indeed think that this may be the case and will be the focus of a follow-up study.*

***Changes made: We have added a discussion on the possible influence of mineral protection on the different TerrOM markers to section 5.2.3.***

**Line 568-569:** For the discussion on priming mechanism:

1) Is there any more detailed evidence of priming, which I do believe is happening in this system. Wysocki et al., 2006 made reference to this which may be useful. I do like the notion of algal-drive material being linked in this ad these materials get processed along the way as they move west. Also, why would priming could enhance the decomposition of soil-derived OM, but not plant-derived OM? This needs some further justification with refs.

*Reply: We thank the reviewer for this suggestion and will add citations to the work of Wysocki et al., to the manuscript. As mentioned earlier, we cannot provide direct evidence that soil-OM is preferentially targeted by priming in this study. However, several previous studies on river transport of brGDGTs (Li et al., 2015 Org. Geochem; Freymond et al., 2017, Org. Geochem.; Kirkels et al., 2022, Biogeosciences) have shown that the proxy signal derived from soil-derived brGDGT represents local environmental conditions and therefore seems to be continuously renewed during transport. Furthermore, these studies have suggested that brGDGTs are not transported in association with mineral surfaces, due to the dissimilar trends in concentration of brGDGTs, bulk OM and elemental compositions of the catchment soils. On the other hand, several studies also report that the brGDGT signal that is discharged is overprinted by brGDGTs that are produced in the coastal marine environment (De Jonge et al., 2014 GCA; Zell et al., 2013 Limnol. Oceanogr; Warden et al., 2016 Biogeosciences; Sinninghe Damsté, 2016, GCA), although we find no evidence for that in our surface sediments. Overall, the composition and transport mechanism of brGDGTs would make these compounds more sensitive to degradation upon discharge. In contrast, a study focusing on the fluvial transport of n-alkanes found that n-alkane concentrations were correlated to fine grain size fractions and aged accordingly with prolonged river transport (Repasch et al., 2022, Geophys. Res. Lett.), indicating that n-alkanes represent a more resistant pool of OM that is likely transported as mineral associated OM. These differences in composition and transport mechanism can possibly result in the preferential targeting of soil OM during priming. As indicated above, a discussion on this topic will be included in the manuscript.*

***Changes made: As also requested by the editor we have substantially extended the discussion on priming (section 5.3 of the manuscript), in which we have addressed all points mentioned above.***

**Line 584-585:** The authors need to better state whether the trends they observed in OM cycling were only controlled by source-differentiation, hydrodynamic transport, and/or hot spots of decomposition.

1) Any proxies here to confirm that the residual of soil-derived OM is more "transformed" than plant-derived OM? Perhaps comparing the concentration of each soil-derived OM biomarker in GoM sediments vs. in riverine sediments. Again, why priming mechanism can facilitate decomposition of soil-derived OM, but not plant-derived OM?

*Reply: With this study, we mainly conclude that source-differentiation causes the differences in OM dispersal patterns as that is what we can firmly conclude from our data-set. Several follow-up studies are being undertaken to assess possible variations in hydrodynamic sorting, transport mechanisms and loss upon discharge (see previous comments). In the summer of 2022 we have collected soil, vegetation material, and riverbed sediments from the Mississippi delta to obtain biomarker and palynological end-members for the terrestrial realm (yes, it was hot). We will use this material to better connect the terrestrial and marine environments.*

**Changes made: We have elaborated on the possible explanations for preferential decomposition of soil (microbial)-derived OM over that of plant-derived OM as part of the discussion in the priming section (5.3).**

3) What's about non-point source input of plant-derived OM (e.g., marshes) vs. soil-derived OM?

*Reply: Non-point material could be derived from aeolian input and/or erosion of sediments from coastal areas. Such input could indeed contribute to the total TerrOM pool in our samples. However, given the amount of water and sediment transported to the coast and offshore by such a big river as the Mississippi, we tend to think that non-point sources are a minor contributor.*
*We do not have a direct way to quantify the total contribution of – for instance - marsh input in our samples, but we can make an estimation by using pollen from typical marsh plants. Our pollen data indicate that marsh taxa occur in higher relative abundances close to shore (~20%), while their total concentration remains overall low. Notably, marsh plant pollen in the shelf sediments can still (also) be introduced by the Mississippi river, as its plume extends westwards onto the shelf. Literature also suggests that the primary source of pollen (and possibly OM) on the Louisiana shelf is the Mississippi river (e.g. Chmura et al., 1999, Paleogeogr.). Furthermore, the average chain length (ACL) of long-chain n-alkanes does not show a spatial trend that would reveal a change in plant type source. The input of e.g. n-alkanes by aeolian transport has been discussed in the manuscript. Therefore, we can conclude that if there is a non-point source of plant input, it is likely neglectable in comparison with the point source represented by the rivers. This will also be clarified in the revised manuscript.*

**Changes made: We added this reasoning to section 5.2.3.**

**Reviewer 2**

My only concern is about the apparent lack of a scientific question. The issue of the fate of terrestrial OM at the land-sea interface is, of course, important for C cycling, but the introduction could indicate why and how the present results will contribute to create new knowledges. Assumptions are formulated to discuss the relative preservation of the different sources of terrestrial OM, then there is just to make this point explicit.
Finally, in order to be more efficient on the use of money and energy, I suggest for further research that, at this stage of the knowledges on the northern Gulf of Mexico system, it could

be interesting to test hypotheses on the fate of OM with a specific sampling strategy or using experimentations instead of formulating assumptions after the sampling.

*Reply:*
*-Research question: In this study, we aim to identify the TerrOM composition stored in the northern GoM to assess if specific OM pools are preferentially buried in the sediments. To further motivate our aim, we will add more information on previous biomarker work in the GoM to the introduction and specify the knowledge gaps that remain. We will also further clarify how our dataset will contribute to closing this knowledge gap.*
*- Study design: Studying the dispersal of TerrOM in the northern GoM was the main objective when planning the research expedition. We have deliberately collected sediments along two land-sea transect starting at the river mouths of the AR and the MR, as well as one shelf transect that follows the MAR river plume to trace TerrOM. We will clarify this further in the revised manuscript.*

**Changes made: We added a more extensive introduction of previous biomarker work in the GoM to the introduction and have made our research aims more explicit, as also requested by the editor.**

---

## Author Response (AR2)

Utrecht, 17 January 2023

Dear editor, dear Ji-Hyung Park,

Thank you for evaluating our revised manuscript and our comments on your and the referees' comments.

Here we resubmit the manuscript titled 'The dispersal of fluvially discharged and marine, shelf-produced particulate organic matter in the northern Gulf of Mexico' by Y.W. Yedema, F. Sangiorgi, A. Sluijs, J.S. Sinninghe Damsté and F. Peterse.

We have revised the manuscript based on the additional suggestions by the reviewer. A point-to-point list of detailed changes in response to comments of the referee can be found below. The changes in the revised manuscript are made with track changes on.

We hope that you find this revised version suitable for publication in Biogeosciences.

On behalf of all co-authors,
Yord Yedema

**Reviewer #1**

- Page 3: Response to Reviewer#1 comment 1, also line 235-245 in the revised manuscript
Sedimentation rate generates by 210Pb data does not really capture seasonal variation of surface sediment properties. Nevertheless, I would suggest the authors add a few sentences that discuss sedimentation and particle dynamics from previous work that use short-lived radionuclides such as Be-7 and Th-234 that can capture seasonal variation (e.g., Corbett, McKee, Allison).

*Reply: We agree that Be-7 and Th-234 radionuclides can capture variations on a smaller scale than Pb-210 data could. Unfortunately, we were unable to measure these short-lived radionuclides, as the our samples were unavailable for a few months due to the COVID pandemic. We have added a few sentences on the seasonal and annual variations in sedimentation rates and particle dynamics to section 3.2, lines 235-242:*

*"Therefore, the upper 0-2 cm sediment analysed, which represents at least 1 year deposition time, integrates multiple years of  seasonally varying oxygen conditions. Still, sediment accumulation rates can vary substantially on a seasonal scale, depending on river discharge, biological processes, and hydrological factors like waves and tides (McKee et al., 2004). For example, sediment accumulation rates near the MR delta derived from short-lived radionuclides ($^{7}$Be and $^{234}$Th) have shown annual and seasonal variations that are larger than the average decadal sedimentation rates calculated with $^{210}$Pb (Corbett et al., 2004), suggesting that active sediment reworking and possible export of particles off- and along-shore takes place. Nevertheless, albeit not completely correct, we here  assumption assume is that such that the integrated, multiple years signal in our surface sediments reduces any the OM burial and preservation biases among locations in our set of GoM sediments."*

- Page 4: Response to Reviewer#1 comment 2

Many publications provide evidences to support that mineral-associated OM is dominated by plant-derived OM (e.g., Angst et al., 2017) while the other showed that microbial-derived OM is more dominant (e.g., Cotrufo et al., 2019; see reviews on this in debate in Lavelle et al., 2019, Angst et al., 2021).

Angst, G., Mueller, K. E., Kögel-Knabner, I., Freeman, K. H., & Mueller, C. W. (2017). Aggregation controls the stability of lignin and lipids in clay-sized particulate and mineral associated organic matter. Biogeochemistry, 132(3), 307-324.
Cotrufo, M. F., Ranalli, M. G., Haddix, M. L., Six, J., & Lugato, E. (2019). Soil carbon storage informed by particulate and mineral-associated organic matter. Nature Geoscience, 12(12), 989-994.
Lavallee, J. M., Soong, J. L., & Cotrufo, M. F. (2020). Conceptualizing soil organic matter into particulate and mineral-associated forms to address global change in the 21st century. Global Change Biology, 26(1), 261-273.
Angst, G., Mueller, K. E., Nierop, K. G., & Simpson, M. J. (2021). Plant-or microbial-derived? A review on the molecular composition of stabilized soil organic matter. Soil Biology and Biochemistry, 156, 108189.

*Reply: We thank the reviewer for their suggestions. We have included information on the association of plant and microbial OM to mineral surfaces in our discussion (end of section 5.2.3). Note that the order of this part has been slightly adjusted, so that we now both discuss the influence of natural resistance of our biomarkers to degradation and protection due to sorption to mineral surfaces on their dispersal in the marine realm.*

- Page 12: Response to section 5.2.3), also line 530-532 in revised manuscript
The explanation in author's response is very clear and reasonable. However, the discussion paragraph from line 530-532 is quite short and did not explain well why the authors needed to examine marsh pollen or changes in ACL. I'd suggest the authors to copy the paragraph from their response and paste it replacing lines 530-532.

*Reply: We agree that the revised manuscript was still brief on this topic and have extended this discussion using parts of our reply letter (Lines 524-529).*

- Line 563-568 in revised manuscript:
What is the primary factor that control how far each organic compound could be transported offshore? Their resistance toward degradation, or how tight they were associated with sediment particles? There are many publications discussed that lignin did actually decay in similar rate compared to the simpler organic molecules (see reviews in Dynarski et al., 2020 and references therein); however, their resistance toward degradation might be controlled by degree of protection by minerals. Maybe say that in this study, lignin and n-alkanes were protected in more stable aggregates formed in soil continuum before being transported to rivers and exported to the ocean. However, diols and sterols were not protected by aggregates and were lost through transportation.

Dynarski, K. A., Bossio, D. A., & Scow, K. M. (2020). Dynamic stability of soil carbon: reassessing the "permanence" of soil carbon sequestration. Frontiers in Environmental Science, 8, 514701.

*Reply: We agree with the reviewer that mineral associations may play an important role in the offshore transport of the different biomarkers. However, our current dataset does not allow us to identify whether the transport of TerrOM is controlled by mineral protection or by its resistance against degradation. Hence, both processes are explained in the discussion, which we have extended to address this additional comment. See section 5.2.3, lines 551-578.*